# Interpretable (meta)factorization of clinical questionnaires to identify general dimensions of psychopathology

## Abstract

Psychiatry research aims at understanding manifestations of psychopathology in behavior, in terms of a small number of latent constructs. These are usually inferred from questionnaire data using factor analysis. The resulting factors and relationship to the original questions are not necessarily interpretable. Furthermore, this approach does not provide a way to separate the effect of confounds from those of constructs, and requires explicit imputation for missing data. Finally, there is no clear way to integrate multiple sets of constructs estimated from different questionnaires. An important question is whether there is a universal, compact set of constructs that would span all the psychopathology issues listed across those questionnaires. We propose a new matrix factorization method designed for questionnaires aimed at promoting interpretability, through bound and sparsity constraints. We provide an optimization procedure with theoretical convergence guarantees, and validate automated methods to detect latent dimensionality on synthetic data. We first demonstrate the method on a commonly used general-purpose questionnaire. We then show it can be used to extract a broad set of 15 psychopathology factors spanning 21 questionnaires from the Healthy Brain Network study. We show that our method preserves diagnostic information against competing methods, even as it imposes more constraints. Finally, we demonstrate that it can be used for defining a short, general questionnaire that allows recovery of those 15 meta-factors, using data more efficiently than other methods.

## 1 Introduction

Standardized questionnaires are a common tool in psychiatric practice and research, for purposes ranging from screening to diagnosis or quantification of severity. A typical questionnaire comprises questions – usually referred to as *items* – reflecting the degree to which particular symptoms or behavioural issues are present in study participants. Items are chosen as evidence for the presence of *latent constructs* giving rise to the psychiatric problems observed. For many common disorders, there is a practical consensus on constructs. If so, a questionnaire may be organized so that subsets of the items can be added up to yield a *subscale score* quantifying the presence of their respective construct. Otherwise, the goal may be to discover constructs through factor analysis.

The *factor analysis* of a questionnaire matrix (#participants × #items) expresses it as the product of a factor matrix (#participants × #factors) and a loading matrix (#factors × #items). The method assumes that answers to items may be correlated, and can therefore be explained in terms of a smaller number of factors. The method yields two real-valued matrices, with uncorrelated columns in the factor matrix. The number of factors needs to be specified a priori, or estimated from data. This solution is often subjected to rotation so that, after transformation, each factor has non-zero loadings on few variables, and each variable has a high-loading on a single factor, if possible. The values of the factors for each participant can then be viewed as a succinct representation of them.

Interpreting what construct a factor may represent is done by considering its loadings over all the items. Ideally, if very few items have a non-zero loading, it will be easy to associate the factor with them. However, in practice, the loadings could be an arbitrary linear combination of items, with positive and negative weights. Factors are real-valued, and neither their magnitude nor their sign are

intrinsically meaningful. Beyond this, any missing data will have to be imputed, or the respective items ommitted, before factor analysis can be used. Finally, patterns in answers that are driven by other characteristics of participants (e.g. age or sex) are absorbed into factors themselves, acting as confounders, instead of being represented separately or controlled for.

Over time, many different questionnaires have been developed. Some focus on constructs relevant to particular disorders or behavioral issues; others aim at screening for a wide range of problems. One important question for psychiatry researchers is how many constructs would suffice to explain most manifestations of psychopathology. In addition to its scientific interest, an answer to this question would also be of clinical use, informing design of light-weight questionnaires designed to estimate all key constructs from a minimal number of items. The availability of datasets such as Healthy Brain Network (Alexander et al., 2017), where tens of questionnaires are collected for thousands of children and adolescent participants, makes it possible to address this question in a data-driven way. However, a joint factor analysis of many questionnaires faces additional obstacles, e.g. their having different response scales, very disparate numbers of items, or patterns of missing entries.

In this paper, we propose to address all of the issues above with a novel matrix factorization method specifically designed for use with questionnaire data, through the following contributions.

**Contribution #1: We introduce Interpretability-Constrained Questionnaire Factorization (ICQF), a new matrix factorization method for questionnaire data.** Our method was designed to incorporate characteristics that increased interpretability of the resulting factors, based on several desiderata from active clinical researchers in psychiatry. First, factor values are constrained to be in the range $[0, 1]$, so as to represent a *degree of presence* of the factor. Second, the loadings across items for each factor have to be in the same range as answers in the original questionnaire (typically, $[0, \texttt{max}]$). This makes it possible to examine them as a *pattern of answers* associated with the factor. Third, the reconstructed matrix obtained by multiplying factors by factor loadings is constrained, so that no entry exceeds the range – or observed maximum value – of the original questionnaire. Fourth, the method handles missing data directly, so no imputation is required. Finally, the method supports pre-specifying some factors to model known variables, such as age or sex, to capture the answer patterns correlated with them (e.g. drinking problems appearing as age increases). We demonstrate ICQF in the Childhood Behavior Checklist (CBCL), a widely used questionnaire, and show that it preserves all diagnostic information in various questionnaires, even with additional regularization.

**Contribution #2: We provide theoretical guarantees on the convergence and performance of the optimization procedure.** We introduce an optimization procedure for ICQF, using alternating minimization with ADMM. We demonstrate that this procedure converges to a local minimum of the optimization problem. We implement blockwise-cross-validation (BCV) to determine the number of factors. If this number of factors is close to that underlying the data, the solution will be close to a global minimum. Finally, we show that our procedure detects the number of factors more precisely than competing methods, as evaluated in synthetic data with different noise density.

**Contribution #3: We use a two-level meta-factorization of 21 questionnaires to identify 15 general factors of psychopathology in children and adolescents.** We apply ICQF individually to 21 Healthy Brain Network questionnaires (first-level), and then again to a concatenation of the resulting 21 factor matrices (second-level), yielding a meta-factorization with 15 interpretable meta-factors. We show that these meta-factors can outperform individual questionnaires in diagnostic prediction. We also show that the meta-factorization can be used to produce a short, general questionnaire, with little loss of diagnostic information, using data much more efficiently than competing methods.

## 2 RELATED WORK

The extraction of latent variables (a.k.a. factors) from matrix data is often done through low rank matrix factorizations, such as singular value decomposition (SVD), principal component analysis (PCA) and exploratory Factor Analysis (hereafter, just FA) (Golub & Van Loan, 2013; Bishop & Nasrabadi, 2006). While SVD and PCA aim at reconstructing the data, FA aims at explaining correlations between (questions) items through latent factors (Bandalos & Boehm-Kaufman, 2010). Factor rotation (Browne, 2001; Sass & Schmitt, 2010; Schmitt & Sass, 2011) is then performed to obtain a sparser solution which is easier to interpret and analyze. For a comprehensive review of FA, see Thompson (2004); Gaskin & Happell (2014); Gorsuch (2014); Goretzko et al. (2021).

Non-negative matrix factorization (NMF) was proposed as a way of identifying sparser, more interpretable latent variables, which can be added to reconstruct the data matrix. It was introduced in Paatero & Tapper (1994) and was further developed in Lee & Seung (2000). Different varieties of NMF-based models have been proposed for various applications, such as the sparsity-controlled (Eggert & Korner, 2004; Qian et al., 2011), manifold-regularized (Lu et al., 2012), orthogonal Ding et al. (2006); Choi (2008), convex/semi-convex (Ding et al., 2008), or archetypal regularized NMF (Javadi & Montanari, 2020). Recently, the Deep-NMF (Trigeorgis et al., 2016; Zhao et al., 2017) and Deep-MF (Xue et al., 2017; Fan & Cheng, 2018; Arora et al., 2019) have been introduced that can model non-linearities on top of (non-negative) factors, when the sample is large (Fan, 2021). These methods do not directly model either the interpretability characteristics or the constraints that we view as desirable. If the goal is to identify latent variables relevant for multiple matrices, the standard approach is multi-view learning (Sun et al., 2019), or variants that can handle only partial overlap in participants across matrices (Ding et al., 2014; Gunasekar et al., 2015; Gaynanova & Li, 2019). Finally, non-negative matrix tri-factorization (NMTF) (Li et al., 2009; Pei et al., 2015), supports an additional matrix mapping between latent representations for different matrices.

Obtaining a factorization with these methods requires both specifying the number of latent variables, and solving an optimization problem. In SVD/PCA, the number of variables is often selected based on the percentage of variance explained, or determined via techniques such as spectral analysis, the Laplace-PCA method, or Velicer's MAP test (Velicer, 1976; Velicer et al., 2000; Minka, 2000). For FA, several methods have been proposed: Bartlett's test (Bartlett, 1950), parallel analysis (Horn, 1965; Hayton et al., 2004), MAP test and comparison data (Ruscio & Roche, 2012). For NMF, iterative detection algorithms are recommended, e.g. the Bayesian information criterion (BIC) (Stoica & Selen, 2004), cophenetic correlation coefficient (CCC) (Fogel et al., 2007) and the dispersion (Brunet et al., 2004). More recent proposals for NMF are Bi-cross-validation (BiCV) (Owen & Perry, 2009) and its generalization, the blockwise-cross-validation (BCV) (Kanagal & Sindhwani, 2010), which we use in this paper. The optimization problem for NMF is non-convex, and different algorithms for solving it have been proposed. Multiplicative update (MU) (Lee & Seung, 2000) is the simplest and mostly used. Projected gradient algorithms such as the block coordinate descent (Cichocki & Phan, 2009; Xu & Yin, 2013; Kim et al., 2014) and the alternating optimization (Kim & Park, 2008; Mairal et al., 2010) aim at scalability and efficiency in larger matrices. Given that our optimization problem has various constraints, we use a combination of alternative optimization and Alternating Direction Method of Multipliers (ADMM) (Boyd et al., 2011; Huang et al., 2016).

## 3 METHODS

### 3.1 INTERPRETABLE CONSTRAINED QUESTIONNAIRE FACTORIZATION (ICQF)

**Inputs** Our method operates on a questionnaire data matrix $M \in \mathbb{R}_{\geq 0}^{n \times m}$ with $n$ participants and $m$ questions, where entry $(i, j)$ is the answer given by participant $i$ to question $j$. Given that questionnaires often have missing data, we also have a mask matrix $\mathcal{M} \in \{0, 1\}^{n \times m}$ of the same dimensionality as $M$, indicating whether each entry is available (=1) or not (=0). Optionally, we may have a confounder matrix $C \in \mathbb{R}_{\geq 0}^{n \times c}$, encoding $c$ known variables for each participant that could account for correlations across questions (e.g. age or sex). If the $j^{\text{th}}$ confound $C_{[:,j]}$ is categorical, we convert it to indicator columns for each value. If it is continuous, we first rescale it into $[0, 1]$ (range in the dataset), and replace it with two new columns, $C_{[:,j]}$ and $1 - C_{[:,j]}$. This mirroring procedure ensures that both directions of the confounding variables are under consideration (e.g. answer patterns more common the younger or the older the participants are).

**Optimization problem** We seek to factorize the questionnaire matrix $M$ as the product of a $n \times k$ factor matrix $W \in [0, 1]$, with the confound matrix $C \in [0, 1]$ as optional additional columns, and a $m \times (k + c)$ loading matrix $Q := [{}^R Q, {}^C Q]$, with a loading pattern ${}^R Q$ over $m$ questions for each of the $k$ factors (and ${}^C Q$ for optional confounds). Denoting the Hadamard product as $\odot$, our optimization problem minimizes the squared error of this factorization

$$
\begin{aligned}
\underset{W \in \mathcal{W}, Q \in \mathcal{Q}, Z \in \mathcal{Z}}{\text{minimize}} \quad & 1/2 \left\| \mathcal{M} \odot (M - Z) \right\|_F^2 + \beta \cdot R(W, Q) \\
\text{such that} \quad & [W, C] Q^T = Z, \ \mathcal{Z} = \{Z | \min(M) \leq Z_{ij} \leq \max(M)\}, \\
& \mathcal{Q} = \{Q | 0 \leq Q_{ij}\} \text{ and } \mathcal{W} = \{W | 0 \leq W_{ij} \leq 1\}
\end{aligned}
\quad \text{(ICQF)}
$$

subject to entries of $\boldsymbol{Q}$ being in the same value range as question answers, so loadings are interpretable, and bounding the reconstruction by the range of values in the questionnaire matrix $\boldsymbol{M}$. We further regularize $\boldsymbol{W}$ and $\boldsymbol{Q}$ through $R(\boldsymbol{W}, \boldsymbol{Q}) := \|\boldsymbol{W}\|_{p,q} + \gamma\|\boldsymbol{Q}\|_{p,q}$, $\gamma = \frac{n}{m}\max(\boldsymbol{M})$, where $\|\boldsymbol{A}\|_{p,q} := (\sum_{i=1}^{m}(\sum_{j=1}^{n}|\boldsymbol{A}_{ij}|^p)^{q/p})^{1/q}$. Here, we use $p = q = 1$ for sparsity control. $\gamma$ is a heuristic to balance the sparsity control between $\boldsymbol{W}$ and $\boldsymbol{Q}$. With a slight abuse of notation, $\gamma$ is absorbed into $\beta$ of $\boldsymbol{Q}$ if no ambiguity results.

**Choice of number of factors** For each $\beta$, we choose the number of factors $k$ using blockwise-cross-validation (BCV). Given a matrix $\boldsymbol{M}$, for each $k$, we shuffle the rows and columns of $\boldsymbol{M}$ and subdivide it into $b_r \times b_c$ blocks. These blocks are split into 10 folds and we repeatedly omit blocks in a fold, factorize the remainder, impute the omitted blocks via matrix completion and compute the error[1] of that imputation. We choose $k$ with the lowest average error. This procedure can adapt to the distribution of confounds $\boldsymbol{C}$ by stratified splitting. We compared this with other approaches for choosing $k$, for ICQF and other methods, over synthetic data, and report the results in Appendix F.

## 3.2 SOLVING THE OPTIMIZATION PROBLEM

**Optimization procedure** The ICQF problem is non-convex and requires satisfying multiple constraints. We solve it through an ADMM optimization procedure. The Lagrangian $\mathcal{L}_\rho$ is:

$$\mathcal{L}_\rho(\boldsymbol{W}, \boldsymbol{Q}, \boldsymbol{Z}, \alpha_{\boldsymbol{Z}}) = 1/2\|\mathcal{M} \odot (\boldsymbol{M} - \boldsymbol{Z})\|_F^2 + \mathcal{I}_{\mathcal{W}}(\boldsymbol{W}) + \beta\|\boldsymbol{W}\|_{1,1} + \mathcal{I}_{\mathcal{Q}}(\boldsymbol{Q}) + \beta\|\boldsymbol{Q}\|_{1,1}$$
$$+ \langle \alpha_{\boldsymbol{Z}}, \boldsymbol{Z} - [\boldsymbol{W}, \boldsymbol{C}]\boldsymbol{Q}^T \rangle + \rho/2\left\|\boldsymbol{Z} - [\boldsymbol{W}, \boldsymbol{C}]\boldsymbol{Q}^T\right\|_F^2 + \mathcal{I}_{\mathcal{Z}}(\boldsymbol{Z}) \tag{1}$$

where $\rho$ is the penalty parameter, $\alpha_{\boldsymbol{Z}}$ is the vector of Lagrangian multipliers and $\mathcal{I}_{\mathcal{X}}(\boldsymbol{X}) = 0$ if $\boldsymbol{X} \in \mathcal{X}$ and $\infty$ otherwise. We alternatingly update primal variables $\boldsymbol{W}, \boldsymbol{Q}$ and the auxiliary variable $\boldsymbol{Z}$ by solving the following sub-problems:

$$\boldsymbol{W}^{(i+1)} = \underset{\boldsymbol{W} \in \mathcal{W}}{\arg\min}\, \rho/2\|\boldsymbol{Z}^{(i)} - [\boldsymbol{W}, \boldsymbol{C}]\boldsymbol{Q}^{(i),T} + \rho^{-1}\alpha_{\boldsymbol{Z}}^{(i)}\|_F^2 + \beta\|\boldsymbol{W}\|_{1,1} \tag{2}$$

$$\boldsymbol{Q}^{(i+1)} = \underset{\boldsymbol{Q} \in \mathcal{Q}}{\arg\min}\, \rho/2\|\boldsymbol{Z}^{(i)} - [\boldsymbol{W}^{(i+1)}, \boldsymbol{C}]\boldsymbol{Q}^T + \rho^{-1}\alpha_{\boldsymbol{Z}}^{(i)}\|_F^2 + \beta\|\boldsymbol{Q}\|_{1,1} \tag{3}$$

$$\boldsymbol{Z}^{(i+1)} = \underset{\boldsymbol{Z} \in \mathcal{Z}}{\arg\min}\, \|\mathcal{M} \odot (\boldsymbol{M} - \boldsymbol{Z})\|_F^2 + \rho\|\boldsymbol{Z} - [\boldsymbol{W}^{(i+1)}, \boldsymbol{C}]\boldsymbol{Q}^{(i+1),T} + \rho^{-1}\alpha_{\boldsymbol{Z}}^{(i)}\|_F^2 \tag{4}$$

for some penalty parameter $\rho$. Lastly, $\alpha_{\boldsymbol{Z}}$ is updated via

$$\alpha_{\boldsymbol{Z}}^{(i+1)} \leftarrow \alpha_{\boldsymbol{Z}}^{(i)} + \rho(\boldsymbol{Z}^{(i+1)} - [\boldsymbol{W}^{(i+1)}, \boldsymbol{C}](\boldsymbol{Q}^{(i+1)})^T) \tag{5}$$

Equations 2 and 3 can be further split into row-wise constrained Lasso problems and there is a closed form solution for equation 4. The optimization details are further discussed in Appendix A. Given the flexibility of ADMM, a similar procedure can also be used with other regularizations.

**Convergence of the optimization procedure** In Appendix B, we provide a proof that the constraint $\rho \geq \sqrt{2}$ on the penalty parameter $\rho$ guarantees monotonicity of the optimization procedure, and that it will converge to a local minimum. Integrating this constraint with the adaptive selection of $\rho$ (Xu et al., 2017), we obtain an efficient optimization for ICQF. Furthermore, Bjorck et al. (2021) showed that, if $k = k^*$ of a ground-truth solution $(\boldsymbol{W}^*, \boldsymbol{Q}^*)$ in non-negative matrix factorizations, the error $\|\boldsymbol{M} - \boldsymbol{W}\boldsymbol{Q}^T\|_F^2$ is star-convex towards $(\boldsymbol{W}^*, \boldsymbol{Q}^*)$, and the solution is close to a global minimum. In Appendix C, we show that, if $k \neq k^*$, the relative error between $\boldsymbol{W}^*$ and $\boldsymbol{W}$ increases with $|\sqrt{k/k^*} - 1|$. Inaccurate estimation of $k^*$ thus affects both the interpretability of $(\boldsymbol{W}, \boldsymbol{Q})$ and the convergence to global minima. As reported in Appendix F, BCV is more robust to noise when estimating $k$ than other alternatives, and this is why we use it.

## 3.3 META-FACTORIZATION

ICQF produces interpretable factors for individual questionnaires. As discussed earlier, our second goal is to obtain interpretable factors that explain psychopathology across a range of questionnaires. ICQF can also be used to obtain these *meta-factors*, through a two level-factorization: factorize each

---

[1]Appropriate weighting is multiplied to the error if number of blocks in the last fold is less than others.

individual questionnaire, concatenate their respective factor matrices, and then factorize this matrix. The main obstacle is that each participant may only have answered a subset of the questionnaires available. This is the second reason for including a mask matrix $\mathcal{M}$ in our problem formulation.

In describing our *meta-factorization* procedure, we suppress $C$ and the regularization terms $R$ to simplify the discussion. Let $\{M_i\}_{i=1}^S$ to be the data matrices of $S$ questionnaires with dimensions $\{(n_i, m_i)\}_{i=1}^S$. Note that $\{M_i\}_{i=1}^S$ can be fully, partially or non-overlapped with each other. For dimension consistency, we extend $M_i$ (so as $W_i$, $\mathcal{M}_i$) into $n$ rows by padding rows of zeros for missing participants, where $n \geq n_i \ \forall i = 1, \ldots, S$ denotes the number of unique participants from the $S$ questionnaires. We then introduce mask matrices $\mathcal{E}_i \in \{0,1\}^{n \times m_i} := \mathcal{D}_i \cdot \mathcal{M}_i$, which is composed of the extended $\mathcal{M}_i$ and a diagonal mask matrix $\mathcal{D}_i \in \{0,1\}^{n \times n}$ indicating the availability of participants. Performing matrix factorization for each questionnaire, we obtain $\mathcal{E}_i \odot M_i \approx \mathcal{E}_i \odot (W_i Q_i^T)$ for $i = 1, \ldots, S$. We then concatenate $\{W_i\}_{i=1}^S$ and perform a second level factorization:

$$[\mathcal{D}_1 \cdot W_1, \cdots, \mathcal{D}_S \cdot W_S] \approx [\mathcal{D}_1 \cdot \mathbb{1}_1, \ldots, \mathcal{D}_S \cdot \mathbb{1}_S] \odot \mathbb{W}\mathbb{Q}^T, \tag{6}$$

where $\mathbb{1}_i$ is a 1-matrix of dimension $n \times k_i$ with $k_i$ denoting the number of factors in $W_i$, for $i = 1, \ldots, S$. The columns of $\mathbb{W}$ are the *meta-factors*.

There are alternative approaches for factorizing multiple questionnaires. The most obvious would be to factorize the concatenation of all $\{M_i\}_{i=1}^S$ as $[\mathcal{E}_1 \odot M_1, \ldots, \mathcal{E}_S \odot M_S] \approx WQ^T$. It requires a wider detection range for the best $k$ ($\{1, \ldots, \sum_i^S m_i\}$). This is used for competing methods in our experiments. Moreover, any low-rank matrix completion algorithms could be used for meta-factorization. However, constraining $W$ and $Z$ from each questionnaire is crucial; as discussed in Maisog et al. (2021), and witnessed by us in practice, simple normalization before estimating $k$ for meta-factor $\mathbb{W}$ may induce unpredictable effects. We could also optimize a *meta-objective function*: $\frac{1}{2} \left( \sum_{i=1}^S \alpha_i \|\mathcal{E}_i \odot (M_i - WQ_i^T)\|_F^2 \right)$. This has a smaller range of $k$ in practice, but extra hyper-parameters $\alpha_i$ (relative importance of data matrix $M_i$) are introduced. Finally, we could use *tri-factorization*: $[\mathcal{E}_1 \odot M_1, \ldots, \mathcal{E}_S \odot M_S] \approx WGQ^T$. This did not work well in our case.

## 4 DATA

The Healthy Brain Network (*HBN*) (Alexander et al., 2017) is an ongoing project to create a biobank from New York City area children and adolescents. Data are publicly available, and include psychiatric, behavioral, cognitive, multimodal brain imaging, and genetics. In this work, we use a subset of 21 psychiatric questionnaires about behavioral and emotional problems. They were selected by domain experts by their focus on psychopathology, frequency of use in clinical research, and completeness in the *HBN* dataset. This subset contains general-purpose questionnaires covering different domains of psychopathology (e.g. *CBCL*, *SDQ* and *SympChck*) and others focusing on specific disorders (e.g. *ASSQ* for autism screening, *SWAN* for ADHD, and *SCARED* for anxiety). The full list of 21 questionnaires is reported in Table 4 in Appendix G. Across all questionnaires, we have 978 questions and 3572 unique participants. Finally, we have the age and sex at birth of each participant, which will be used as confounds, and diagnostic labels for 11 conditions, if applicable.

## 5 EXPERIMENTS AND RESULTS

### 5.1 ICQF FACTORIZATION OF THE CHILD BEHAVIOR CHECKLIST

We begin with a qualitative assessment of ICQF applied to the 2001 Child Behavior Checklist (*CBCL*), which is designed to detect behavioral issues. The checklist includes 113 questions, grouped into 8 syndrome subscales: *Aggressive, Anxiety/Depressed, Attention, Rule Break, Social, Somatic, Thought, Withdrawn* problems. Answers are scored on a three-point Likert scale (0=absent, 1=occurs sometimes, 2=occurs often) and the time frame for the responses is the past 6 months.

We estimated the latent dimensionality $k = 8$ using BCV to compute a test error for ICQF at each possible $k$. The regularization parameter $\beta = 0.5$ was set the same way (See bottom-left-panel of Figure 1). The top-panel shows the heat map of $Q := [^RQ, {}^CQ]$, the loadings over questions $^RQ$ for the latent factors $W$, and the loadings $^CQ$ for the confounds $C$. Questions are grouped by syndrome subscale. While there were factors that loaded primarily in questions from one subscale, as expected,

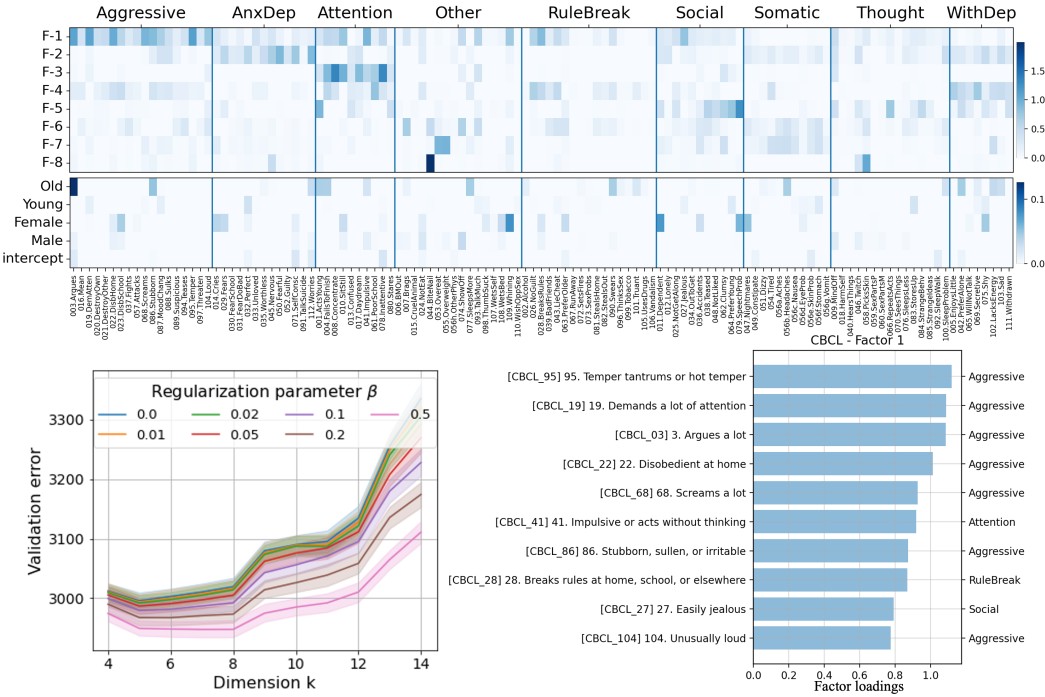

Figure 1: **Top:** Heat map of factor and confound loadings $\boldsymbol{Q} := [{}^R\boldsymbol{Q}, {}^C\boldsymbol{Q}]$. Note that questions are grouped by syndrome subscale; some factors are syndrome specific, while others bridge syndromes. **Bottom left:** Detection of $(k, \beta)$ by BCV. **Bottom right:** Top 10 questions of $\boldsymbol{Q}_{[:,1]}$, loading mostly on the *Aggressive* subscale. Top 10 questions for all factors are shown in Appendix H.

we were surprised to find others that grouped questions from multiple subscales. These were deemed sensible co-occurrences by our clinical collaborators. We show the top 10 questions, ranked by magnitude of loading, for the first factor $\boldsymbol{Q}_{[:,1]}$ from ${}^R\boldsymbol{Q}$, as a demonstration of how one might interpret the factor (bottom-right-panel of Figure 1). As a further, sanity check, we inspected the loadings of confound **Old** (increasing age) and observe that they covered issues such as *"Argues", "Act Young", "Swears" and "Alcohol"*. The loadings of $\boldsymbol{Q}$ also reveal the relative importance among questions in each estimated factor; subscales deem all questions equally important.

## 5.2 META-ICQF FOR META-FACTORIZATION OF THE HBN QUESTIONNAIRES

This section provides a qualitative evaluation of meta-ICQF, analogous to that of ICQF in Section 5.1. As described earlier, the meta-factorization requires a first-level ICQF of each questionnaire in *HBN*, yielding factor matrix $\boldsymbol{W}_i$ and loading matrix $\boldsymbol{Q}_i$. On the second level factorization, we concatenate $\{\boldsymbol{W}_i\}_{i=1}^S$ and use ICQF to get meta-factors $\mathbb{W}$ and respective loadings $\mathbb{Q}$ over first level factors as described in equation 6 (note the change in font for these). Figure 2 (left) shows the lower triangular part of the correlation matrix of $\mathbb{Q}^T$. The first level factors from all questionnaires are grouped through agglomerative clustering – as many clusters as meta-factors $k$ – on their meta-factor loadings. The sparse, block-diagonal pattern and the diversified factor-origins within each block demonstrate how meta-factorization can combine related latent factors from multiple questionnaires. Figure 2 (bottom-right) shows the trend of validation errors with different $(k, \beta)$ using the BCV detection scheme. The optimal inflection point is $k = 15$ and $\beta = 0.1$. Finally, we can back-propagate $\mathbb{Q}$ from factor- to question-level[2] by multiplying $\text{diag}[\boldsymbol{Q}_1, \dots, \boldsymbol{Q}_S] \cdot \mathbb{Q} =: \boldsymbol{Q}$. These loadings retrieve the question's latent representation in the meta-factor space; the magnitude of each question's entry in each column of $\mathbb{Q}$ reveals its influence of the corresponding meta-factor. Figure 2 (top-right) shows the top 10 questions of the first column of $\mathbb{Q}$ ranked by their magnitude,

---

[2]For multi-questionnaire setting, we abuse the notation $\boldsymbol{Q}$ to denote the latent representation of questions, either by direct factorization, or the meta-factorization followed by back-propagation.

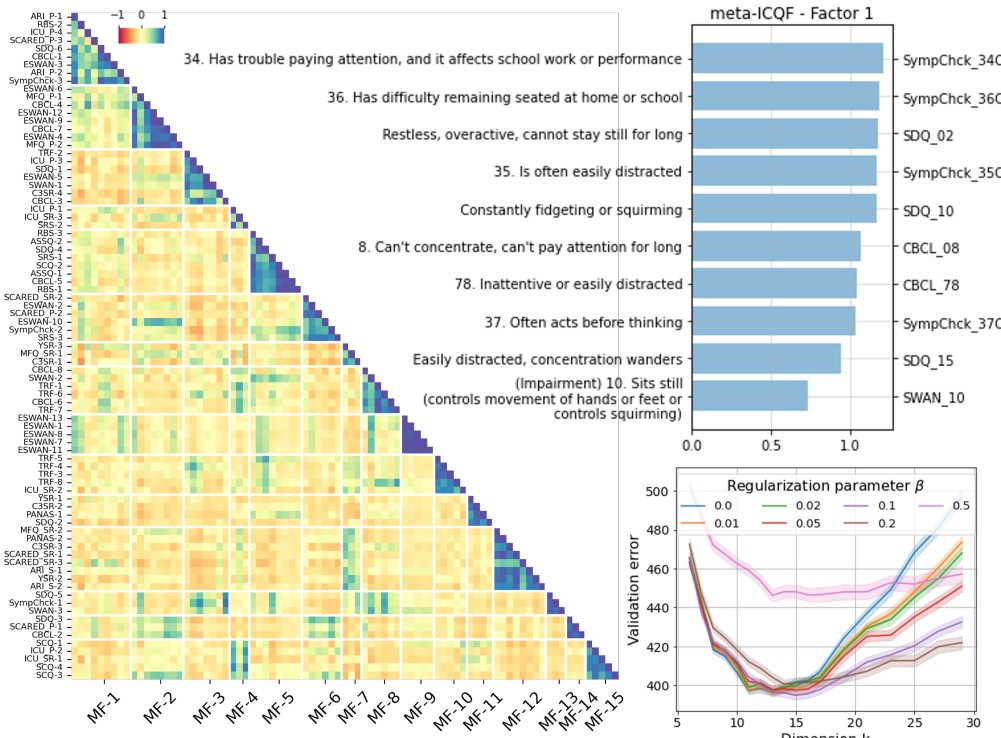

Figure 2: **Left:** Correlation matrix of $\mathbb{Q}$, showing clusters of similar ICQF factors for all questionnaires on the left (format $questionnaire - \#$), and the meta-factors (1-15) those clusters correspond to at the bottom. **Bottom-right:** Detection of $(k, \beta)$ by BCV. **Top-right:** The top 10 questions associated with meta-factor 1, their loading value, and their questionnaire of origin. This meta-factor reflects attention issues. A similar plot for every meta-factor is reported in Appendix I.

with their questionnaire of origin. This meta-factor reflects attention issues. Similar plots for all 15 meta-factors are reported in Appendix I. There is strong topic coherence among the top ranked questions in each meta-factor. The meta-factors have been deemed interpretable and clinically plausible presentations by our psychiatry collaborators.

## 5.3 DIAGNOSTIC CLASSIFICATION

Given the absence of ground-truth factorizations for participants in the *HBN* study, it is challenging to carry out a quantitative evaluation of ICQF versus other factorization methods, or subscales. In this section, we report on two different experiments based on predicting diagnostic labels for each participant, from factor scores. The first tests whether factor matrices $\boldsymbol{W}$ preserve the necessary information for this, when applied to general-purposed questionnaires or a combination of every *HBN* questionnaire. The second tests whether the question ranking induced by $\boldsymbol{Q}$, across all questionnaires in *HBN*, selects the most informative questions for each factor.

### 5.3.1 EXPERIMENTAL SETUP

**Baseline methods** Our first baseline method is $\ell_1$- regularized NMF ($\ell_1$-NMF) (Cichocki & Phan, 2009), as it also imposes non-negativity and sparsity constraints. As constructs (or questions) can be correlated, we rule out other NMF methods with orthogonality constraints. FA with promax rotation (FA-promax) (Hendrickson & White, 1964) using minimum residual as estimation method is included because it is commonly used in analyzing questionnaires. Syndrome subscales are included if available for a questionnaire, since they are often used for diagnoses. Finally, we include raw questionnaire answers, as they have all the information available. To estimate the number of factors $k$, we use BCV for $\ell_1$-NMF and ICQF, and parallel analysis for FA. The choice was driven by the experiments on synthetic questionnaire data reported in Appendix F.

Table 1: Averaged ROC-AUC scores of the diagnostic prediction under different factorizations.

| Questionnaire | Factorization | | | | | |
|---|---|---|---|---|---|---|
| | meta-ICQF | ICQF | $\ell_1$-NMF | FA-promax | subscales | raw |
| CBCL | – | 0.768 | 0.762 | 0.763 | 0.740 | 0.763 |
| SDQ | – | 0.762 | 0.755 | 0.757 | 0.752 | 0.739 |
| SympChck | – | 0.753 | 0.761 | 0.756 | NaN | 0.747 |
| HBN | 0.792 | 0.782 | 0.777 | 0.778 | 0.766 | 0.788 |

**Questionnaires** The two experiments are motivated by the routine use of general-purpose questionnaires in our dataset – *CBCL, SDQ* (Symptoms and Difficulties), and *SympChck* (Symptom check) – to screen and refer patients to pediatric psychiatry clinics, for a variety of diagnoses (Heflinger et al., 2000; Biederman et al., 2005; 2020). The referral is based either on raw answers on the questionnaire or syndrome-specific subscales derived from them. Beyond this, and given that we have 21 questionnaires from *HBN*, we carried out experiments on $\mathbb{W}$ derived from them. The factors are obtained using the meta-factorization described in Section 3.3 (meta-ICQF), or by concatenating the questionnaires and factorizing the result ($\ell_1$-NMF, FA-promax), or simply using the concatenation (raw), possibly aggregated (by subscales, if defined, or all added otherwise).

**Dataset splits** We use a similar evaluation procedure in both experiments. We group the 21 *HBN* questionnaires, and split participants into train, validation, and test sets with ratio $70/15/15$, based on participant availability across questionnaires and the distribution of confounds and diagnostic labels. This ensures a similar data distribution in the three sets, as shown in Figure 5 in Appendix E, where more details are provided. We resample 50 dataset splits using different seeds, and carry out both experiments in each split. The results reported are the average across results in all splits.

**Model training and inference** Let $W_i^{set}$ denote the participant factor matrix in ICQF or NMF, or the factor score in FA. The subscript $i$ is the questionnaire index, and is dropped if considering only one. The superscript denotes the set. Similarly, let $Q_i$ denote the question loadings for each method. Model training will yield a $(W^{train}, Q)$ for participants in the training set. Inference with the model will produce $W^{validate}$ and $W^{test}$ in validation and test sets, using the trained $Q$ and confounds $C^{validate}, C^{test}$ (if applicable). See Figure 4 in Appendix D for a diagram. A sans serif font (e.g. $\mathbb{W}$) indicates a second-level factorization result. In meta-factorization, both model training and inference are performed on individual questionnaires to obtain $\{W_i^{train}, W_i^{validate}, W_i^{test}, Q_i\}_{i=1}^S$ at the first level. At the second level, we concatenate $\{W_i^{train}\}_{i=1}^S$ and do model training to get meta-factor $\mathbb{W}^{train}$ and $\mathbb{Q}$, followed by model inference on the concatenated $\{W_i^{validate}\}_{i=1}^S$ and $\{W_i^{test}\}_{i=1}^S$. This then gives $\mathbb{W}^{validate}$ and $\mathbb{W}^{test}$. While meta-factorization is possible for NMF or FA, we do not use it, as results were same or worse than factorizing concatenated questionnaires.

### 5.3.2 DIAGNOSTIC PREDICTION FROM FACTORS

For each one of 11 diagnostic labels, we train a logistic regression model with $\ell_2$ regularization, and balanced class weights, on $W^{train}$ ($\mathbb{W}^{train}$). The regularization strength is tuned using $W^{validate}$ ($\mathbb{W}^{validate}$). Prediction assessment is conducted on $W^{test}$ ($\mathbb{W}^{test}$) using the ROC-AUC metric (Krzanowski & Hand, 2009). The ROC-AUC of each setting is then averaged across all random dataset splits. As results were obtained over the same datasets, for 11 different classification problems, we use a Friedman test with significance level $\alpha = 0.05$ (followed by a posthoc Nemenyi test if the null hypothesis of the Friedman test is rejected), following Demšar (2006).

Table 1 shows the summary of AUCs obtained on the various questionnaires using different factorizations, averaged across all 11 diagnostic labels. *HBN* corresponds to the use of all 21 questionnaires, as described earlier. Results for each problem are provided in Table 5 in Appendix J. In *CBCL*, the null hypothesis is rejected and the post-hoc Nemenyi test indicates that *subscales* is significantly worse than all other factorizations. The null is not rejected for *SDQ* or *SympChck*. In the *HBN* setting, the null hypothesis is rejected and the Nemenyi test indicates a significant difference between the group (meta-ICQF, raw) and the group ($\ell_1$-NMF, subscales). Overall, we conclude that ICQF and meta-ICQF preserve diagnostic information, in spite of additional regularization and constraints versus other methods. Human-defined subscales are slightly but significantly worse. A parallel experiment on another *CBCL* dataset from different population is reported in Appendix K.

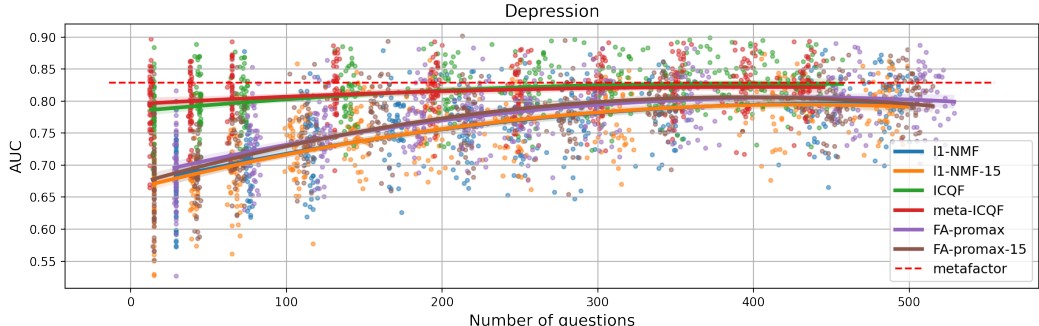

Figure 3: ROC-AUC trend of prediction performance for Depression using the $t$-questionnaire, for increasing $t$. The methods are ICQF (estimated $k = 22$), meta-ICQF (estimated $k = 15$), $\ell_1$-NMF and FA-promax (with $k = 15$ or estimated), and metafactor (meta-ICQF with all questions in *HBN*)

### 5.3.3 DIAGNOSTIC PREDICTION FROM SYNTHESIZED QUESTIONNAIRES

Our goal is to assess the degree to which the process of extracting factors from all of the *HBN* questionnaires yields a $\boldsymbol{Q}$ where loadings identify informative questions for every domain of psychopathology. If so, we should be able to parlay those loadings into general-purpose questionnaires defined in a purely data-driven way, using the most informative question subset. We operationalize this as follows. We first perform factorization in the training set, as described for the previous prediction problem. For each column $\boldsymbol{Q}_{[:,i]}$, we rank questions according to the absolute magnitude of their loadings. By grouping top $t$ questions from each of the $k$ (meta-) factors, we can derive a new questionnaire, which we call the $t$-*questionnaire*. The $t$-*questionnaire* inherits the ranking of questions and ideally, preserves the key information for diagnostic prediction.

The experimental setup parallels Section 5.3.2, but using either the 21 components of the $t$-questionnaire, for meta-ICQF, or their concatenation, otherwise. For the latter we trained $\ell_1$-NMF and FA-promax, either selecting the number of factors from data as before, or setting it to that of meta-ICQF ($k = 15$). We also trained ICQF directly on the concatenation (estimated $k = 22$). We then trained a $\ell_2$ regularized logistic regression, with regularization parameter set over validation data, and evaluated its performance on the test set. This procedure was carried out for $t$ up 40 (or #questions if $< t$). Figure 3 shows the trend of prediction performance for Depression for increasing $t$. The red dotted line is the average performance without eliminating any questions (same as *HBN* in Table 5). The trends for the other 10 diagnostic predictions are reported in Appendix L. They are broadly similar in relative terms across the methods (except for suspected ASD). Across the range of $t$, meta-ICQF and ICQF had substantially higher AUCs than other methods, especially when $t$ was small. This suggests that meta-ICQF is effective at determining the relative importance of $\sim 1000$ questions from 21 questionnaires, as well as grouping them into interpretable meta-factors.

## 6 DISCUSSION

In this paper, we have introduced ICQF, a method for non-negative matrix factorization with additional constraints to further enhance the interpretability of factors, and the capability to directly handle confounds. We have demonstrated ICQF in a widely used questionnaire, and showed that interpretability does not affect our ability to make diagnostic predictions from factors. We also showed that ICQF can be used for two-level meta-factorizations of sets of questionnaires. This allowed us to identify 15 meta-factors of psychopathology that coherently group questions from many questionnaires, and correspond to clinical presentations of patients. Furthermore, we showed that the resulting meta-factorization induces a ranking of the most informative questions for each questionnaire. This makes it possible to generate a minimal general-purpose questionnaire for estimating the 15 meta-factors, while maintaining a specified level of diagnostic prediction performance. In the future, we plan to use the 15 meta-factors as latent variables for studying the structural and functional brain imaging data available in *HBN*. We also plan on releasing the ICQF code for community use, and carrying out clinical validations of the $t$-questionnaires generated from the meta-factorization.

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

APPENDIX

CONTENT

## A OPTIMIZATION PROCEDURE OF ICQF

Recall that the Lagrangian $\mathcal{L}_\rho$ of ICQF is:

$$\mathcal{L}_\rho(\boldsymbol{W}, \boldsymbol{Q}, \boldsymbol{Z}, \alpha_{\boldsymbol{Z}}) = \frac{1}{2}\|\mathcal{M} \odot (\boldsymbol{M} - \boldsymbol{Z})\|_F^2 + \mathcal{I}_{\mathcal{W}}(\boldsymbol{W}) + \beta\|\boldsymbol{W}\|_{1,1} + \mathcal{I}_{\mathcal{Q}}(\boldsymbol{Q}) + \beta\|\boldsymbol{Q}\|_{1,1}$$
$$+ \left\langle \alpha_{\boldsymbol{Z}}, \boldsymbol{Z} - [\boldsymbol{W}, \boldsymbol{C}]\boldsymbol{Q}^T \right\rangle + \frac{\rho}{2}\left\| \boldsymbol{Z} - [\boldsymbol{W}, \boldsymbol{C}]\boldsymbol{Q}^T \right\|_F^2 + \mathcal{I}_{\mathcal{Z}}(\boldsymbol{Z})$$

Following the ADMM approach, we alternatingly update primal variables $\boldsymbol{W}, \boldsymbol{Q}$ and the auxiliary variable $\boldsymbol{Z}$, instead of updating them jointly. In particular, we iteratively solve the following subproblems:

$$\boldsymbol{W}^{(i+1)} = \underset{\boldsymbol{W} \in \mathcal{W}}{\arg\min} \frac{\rho}{2}\left\| \boldsymbol{Z}^{(i)} - [\boldsymbol{W}, \boldsymbol{C}]\boldsymbol{Q}^{(i),T} + \frac{1}{\rho}\alpha_{\boldsymbol{Z}}^{(i)} \right\|_F^2 + \beta\|\boldsymbol{W}\|_{1,1} \qquad \text{(Sub-problem 1)}$$

$$\boldsymbol{Q}^{(i+1)} = \underset{\boldsymbol{Q} \in \mathcal{Q}}{\arg\min} \frac{\rho}{2}\left\| \boldsymbol{Z}^{(i)} - [\boldsymbol{W}^{(i+1)}, \boldsymbol{C}]\boldsymbol{Q}^T + \frac{1}{\rho}\alpha_{\boldsymbol{Z}}^{(i)} \right\|_F^2 + \beta\|\boldsymbol{Q}\|_{1,1} \qquad \text{(Sub-problem 2)}$$

$$\boldsymbol{Z}^{(i+1)} = \underset{\boldsymbol{Z} \in \mathcal{Z}}{\arg\min} \frac{1}{2}\|\mathcal{M} \odot (\boldsymbol{M} - \boldsymbol{Z})\|_F^2 + \frac{\rho}{2}\left\| \boldsymbol{Z} - [\boldsymbol{W}^{(i+1)}, \boldsymbol{C}]\boldsymbol{Q}^{(i+1),T} + \frac{1}{\rho}\alpha_{\boldsymbol{Z}}^{(i)} \right\|_F^2$$
$$\text{(Sub-problem 3)}$$

for some penalty parameter $\rho$. We denote the Hadamard product as $\odot$. The vector of Lagrangian multipliers $\alpha_{\boldsymbol{Z}}$ is updated via

$$\alpha_{\boldsymbol{Z}}^{(i+1)} \leftarrow \alpha_{\boldsymbol{Z}}^{(i)} + \rho(\boldsymbol{Z}^{(i+1)} - [\boldsymbol{W}^{(i+1)}, \boldsymbol{C}](\boldsymbol{Q}^{(i+1)})^T) \qquad (7)$$

SUB-PROBLEMS 1 AND 2 (EQUATIONS 2 AND 3)

Note that equation 2 (and similarly equation 3 by taking the transpose) can be split into row-wise constrained Lasso problem. Specifically, the $r^{\text{th}}$ row problem can be simplified into:

$$\boldsymbol{x}^* = \arg\min_{0 \le \boldsymbol{x}_i \le 1} \frac{\rho}{2} \|\boldsymbol{b} - \boldsymbol{A}\boldsymbol{x}\|_F^2 + \beta \|\boldsymbol{x}\|_1, \quad \boldsymbol{A} = \boldsymbol{Q}^{(i)}, \quad \boldsymbol{b} = \left[ \boldsymbol{Z}^{(i)} - \boldsymbol{C}\boldsymbol{Q}^{(i),T} + \frac{1}{\rho}\alpha_{\boldsymbol{Z}}^{(i)} \right]_{[r,:]} \quad (8)$$

Here we use the Matlab matrix notation $\left[ \cdot \right]_{[r,:]}$ to represent row extraction operation. As suggested in Gaines et al. (2018) one can also use ADMM to solve equation 8:

$$\boldsymbol{x}^{(i+1)} = \arg\min_{\boldsymbol{x}} \frac{\rho}{2} \|\boldsymbol{b} - \boldsymbol{A}\boldsymbol{x}\|_2^2 + \frac{\tau}{2} \|\boldsymbol{x} - \boldsymbol{y}^{(i)} + \frac{1}{\tau}\mu^{(i)}\|_2^2 + \beta \|\boldsymbol{x}\|_1 \quad (9)$$

$$\boldsymbol{y}^{(i+1)} = Proj_{[0,1]}(\boldsymbol{x}^{(i+1)} + \frac{1}{\tau}\mu^{(i)}) \quad (10)$$

$$\mu^{(i+1)} \leftarrow \mu^{(i)} + \tau(\boldsymbol{x}^{(i+1)} - \boldsymbol{y}^{(i+1)}) \quad (11)$$

Similarly, $\mu$ is the vector of Lagrangian multipliers and $\tau$ is the penalty parameter. $Proj_{[0,1]}$ refers to the orthogonal projection into $[0,1]$ (inherited from the box-constraints of $\boldsymbol{W}$). Equation 9 can be solved via the well-established FISTA algorithm (Beck & Teboulle, 2009). Consider the following optimization problem

$$\arg\min_{\boldsymbol{x}} \ \lambda\|\boldsymbol{x}\|_1 + \frac{1}{2}f(\boldsymbol{x}) \quad (12)$$

The FISTA algorithm for solving 12 is summarized as follows:

---
**Algorithm 1:** FISTA for equation 12

---
**Initialize:** $\delta = 1e{-}6$; $\boldsymbol{x}_{-1} = \boldsymbol{0}, \boldsymbol{x}_0 = \boldsymbol{t}_0 = \boldsymbol{1}$
**Input:** $L$, Lipschitz constant of $\nabla f$
**Result:** Solution $\boldsymbol{x}$ of equation 12
**while** $\|\boldsymbol{x}_i - \boldsymbol{x}_{i-1}\|_2 > \delta$ **do**

$\qquad \widetilde{\boldsymbol{x}}_{i+1} = \arg\min_{\boldsymbol{z}} \left\{ \frac{\lambda}{L}\|\boldsymbol{z}\|_1 + \frac{1}{2} \left\| \boldsymbol{z} - \left( \boldsymbol{x}_i - \frac{1}{L}\nabla f(\boldsymbol{x}_i) \right) \right\| \right\}$;

$\qquad \boldsymbol{t}_{i+1} = \frac{1 + \sqrt{1 + 4\boldsymbol{t}_i^2}}{2}$;

$\qquad \boldsymbol{x}_{i+1} = \widetilde{\boldsymbol{x}}_{i+1} + \frac{\boldsymbol{t}_i - 1}{\boldsymbol{t}_{i+1}}(\widetilde{\boldsymbol{x}}_{i+1} - \boldsymbol{x}_i)$;

**end**

---

To solve equation 9 with FISTA algorithm, using the notation as introduced in equation 8, we have

$$f(\boldsymbol{x}) = \rho\|\boldsymbol{b} - \boldsymbol{A}\boldsymbol{x}\|_2^2 + \tau\|\boldsymbol{x} - \boldsymbol{y}^{(i)} + \frac{1}{\tau}\mu^{(i)}\|_2^2 \quad (13)$$

To compute $L$, the Lipschitz constant of $\nabla f$, we have

$$\begin{aligned} \nabla f(\boldsymbol{x}) &= 2\rho \left( \boldsymbol{A}^T\boldsymbol{A}(\boldsymbol{x} - \boldsymbol{b}) + \tau(\boldsymbol{x} - \boldsymbol{c}) \right) \\ &= 2(\rho\boldsymbol{A}^T\boldsymbol{A} + \tau\boldsymbol{I})\boldsymbol{x} - 2(\rho\boldsymbol{A}^T\boldsymbol{A}\boldsymbol{b} + \tau\boldsymbol{c}) \end{aligned} \quad (14)$$

where $\boldsymbol{c} = \boldsymbol{y}^{(i)} - \frac{1}{\tau}\mu^{(i)}$. Thus, $L$ is just equal to the largest eigenvalue of $2(\rho\boldsymbol{A}^T\boldsymbol{A} + \tau\boldsymbol{I})$.

As recommended in Huang et al. (2016), ADMM provides flexibility to use various types of loss functions and regularizations without changing the procedure. For example, we can simply change to $L_{2,1}$ norm and equation 8 becomes a constrained ridge-regression problem, which can be efficiently solved by non-negative quadratic programming algorithms. For most clinical usage, the size of questionnaire data is manageable on a single machine. However, if optimal computational and memory efficiency is required, various stochastic optimization approaches such as Mairal et al. (2010) can replace the ADMM procedure. Yet, an unbiased sampling scheme for generating random batches that handles missing responses is also needed. Such a scheme is non-trivial to obtain, especially under the multi-questionnaires scenario.

SUB-PROBLEM 3 (EQUATION 4)

Since both terms in equation 4 are in Frobenius-norm, $\boldsymbol{Z}$ can be optimized entry-wise. In particular, we have the following closed-form solution for $\boldsymbol{Z}^{(i+1)}$:

$$\boldsymbol{Z}^{(i+1)} = \underset{[\min(\boldsymbol{M}),\max(\boldsymbol{M})]}{Proj} \left( \mathcal{M} \odot \boldsymbol{M} + \rho[\boldsymbol{W}^{(i+1)}, \boldsymbol{C}](\boldsymbol{Q}^{(i+1)})^T - \alpha_{\boldsymbol{Z}}^{(i)} \right) \oslash (\rho\mathbb{1} + \mathcal{M}) \quad (15)$$

where $\mathbb{1}$ is a 1-matrix with appropriate dimension and $\oslash$ is the Hadamard division.

## B  NON-INCREASING PROPERTY OF THE OPTIMIZATION ALGORITHM

In the following, we provide a self-contained convergence proof and show that, under an appropriate choice of the penalty parameter $\rho$, the ADMM optimization scheme discussed in Section 3.2 converges to a local minimum. To simplify notation, we denote $\mathbb{V}^{(i,j,k)} = \{\boldsymbol{W}^{(i)}, \boldsymbol{Q}^{(j)}, \boldsymbol{Z}^{(k)}\}$ to be the tuple of variables $\boldsymbol{W}, \boldsymbol{Q}$ and $\boldsymbol{Z}$ during iteration $(i), (j)$ and $(k)$ respectively. If $i = j = k$, we abbreviate it as $\mathbb{V}^{(i)}$. We also denote $\boldsymbol{R}^{(i)} = [\boldsymbol{W}^{(i)}, \boldsymbol{C}](\boldsymbol{Q}^{(i)})^T$ and for any matrices $\boldsymbol{A}, \boldsymbol{B}$ with appropriate dimensions, $\langle \boldsymbol{A}, \boldsymbol{B} \rangle = \text{Trace}(\boldsymbol{A}^T\boldsymbol{B})$. In the following, we are going to show that the Lagrangian is decreasing across iterations. Particularly, we consider the difference of Lagrangian between consecutive iterations:

$$\mathcal{L}_\rho(\mathbb{V}^{(i+1)}, \alpha_{\boldsymbol{Z}}^{(i+1)}) - \mathcal{L}_\rho(\mathbb{V}^{(i)}, \alpha_{\boldsymbol{Z}}^{(i)})$$
$$= \underbrace{\mathcal{L}_\rho(\mathbb{V}^{(i+1)}, \alpha_{\boldsymbol{Z}}^{(i+1)}) - \mathcal{L}_\rho(\mathbb{V}^{(i+1)}, \alpha_{\boldsymbol{Z}}^{(i)})}_{(I)} + \underbrace{\mathcal{L}_\rho(\mathbb{V}^{(i+1)}, \alpha_{\boldsymbol{Z}}^{(i)}) - \mathcal{L}_\rho(\mathbb{V}^{(i)}, \alpha_{\boldsymbol{Z}}^{(i)})}_{(II)} \quad (16)$$

Expanding term $(I)$, we have

$$\mathcal{L}_\rho(\mathbb{V}^{(i+1)}, \alpha_{\boldsymbol{Z}}^{(i+1)}) - \mathcal{L}_\rho(\mathbb{V}^{(i+1)}, \alpha_{\boldsymbol{Z}}^{(i)}) = \left\langle \alpha_{\boldsymbol{Z}}^{(i+1)} - \alpha_{\boldsymbol{Z}}^{(i)}, \boldsymbol{Z}^{(i+1)} - \boldsymbol{R}^{(i+1)} \right\rangle$$
$$= \frac{1}{\rho}\|\alpha_{\boldsymbol{Z}}^{(i+1)} - \alpha_{\boldsymbol{Z}}^{(i)}\|_F^2 \quad (17)$$

Expanding term $(II)$, we have

$$\mathcal{L}_\rho(\mathbb{V}^{(i+1)}, \alpha_{\boldsymbol{Z}}^{(i)}) - \mathcal{L}_\rho(\mathbb{V}^{(i)}, \alpha_{\boldsymbol{Z}}^{(i)})$$
$$= \overbrace{\mathcal{L}_\rho(\mathbb{V}^{(i+1)}, \alpha_{\boldsymbol{Z}}^{(i)}) - \mathcal{L}_\rho(\mathbb{V}^{(i+1,i+1,i)}, \alpha_{\boldsymbol{Z}}^{(i)})}^{(\mathcal{A})} + \overbrace{\mathcal{L}_\rho(\mathbb{V}^{(i+1,i+1,i)}, \alpha_{\boldsymbol{Z}}^{(i)}) - \mathcal{L}_\rho(\mathbb{V}^{(i+1,i,i)}, \alpha_{\boldsymbol{Z}}^{(i)})}^{(\mathcal{B})}$$
$$+ \underbrace{\mathcal{L}_\rho(\mathbb{V}^{(i+1,i,i)}, \alpha_{\boldsymbol{Z}}^{(i)}) - L(S^{(k)}, \alpha_{\boldsymbol{Z}}^{(i)})}_{(\mathcal{C})} \quad (18)$$

Expanding $(\mathcal{A})$ by the definition, we have

$$\frac{1}{2}\|\mathcal{M} \odot (\boldsymbol{M} - \boldsymbol{Z}^{(i+1)})\|_F^2 - \frac{1}{2}\|\mathcal{M} \odot (\boldsymbol{M} - \boldsymbol{Z}^{(i)})\|_F^2 + \left\langle \alpha_{\boldsymbol{Z}}^{(i)}, \boldsymbol{Z}^{(i+1)} - \boldsymbol{R}^{(i+1)} \right\rangle$$
$$- \left\langle \alpha_{\boldsymbol{Z}}^{(i)}, \boldsymbol{Z}^{(i)} - \boldsymbol{R}^{(i+1)} \right\rangle + \frac{\rho}{2}\left\|\boldsymbol{Z}^{(i+1)} - \boldsymbol{R}^{(i+1)}\right\|_F^2 - \frac{\rho}{2}\left\|\boldsymbol{Z}^{(i)} - \boldsymbol{R}^{(i+1)}\right\|_F^2$$
$$= \left\langle \mathcal{M} \odot (\boldsymbol{Z}^{(i+1)} - \boldsymbol{M}), \mathcal{M} \odot (\boldsymbol{Z}^{(i+1)} - \boldsymbol{Z}^{(i)}) \right\rangle - \|\mathcal{M} \odot (\boldsymbol{Z}^{(i+1)} - \boldsymbol{Z}^{(i)})\|_F^2$$
$$+ \langle \alpha_{\boldsymbol{Z}}^{(i)}, \boldsymbol{Z}^{(i+1)} - \boldsymbol{Z}^{(i)} \rangle + \rho\left\langle \boldsymbol{Z}^{(i+1)} - \boldsymbol{R}^{(i+1)}, \boldsymbol{Z}^{(i+1)} - \boldsymbol{Z}^{(i)} \right\rangle - \rho\|\boldsymbol{Z}^{(i+1)} - \boldsymbol{Z}^{(i)}\|_F^2$$
$$= \left\langle \mathcal{M} \odot (\boldsymbol{Z}^{(i+1)} - \boldsymbol{M}) + \rho \cdot \boldsymbol{Z}^{(i+1)} + \alpha_{\boldsymbol{Z}}^{(i)} - \rho\boldsymbol{R}^{(i+1)}, \boldsymbol{Z}^{(i+1)} - \boldsymbol{Z}^{(i)} \right\rangle$$
$$- \|\mathcal{M} \odot (\boldsymbol{Z}^{(i+1)} - \boldsymbol{Z}^{(i)})\|_F^2 - \rho\|(\boldsymbol{Z}^{(i+1)} - \boldsymbol{Z}^{(i)})\|_F^2$$
$$- \left\langle \mathcal{M} \odot (\boldsymbol{Z}^{(i+1)} - \boldsymbol{M}), (1 - \mathcal{M}) \odot (\boldsymbol{Z}^{(i+1)} - \boldsymbol{Z}^{(i)}) \right\rangle$$

Since $\boldsymbol{Z}^{(i+1)}$ is the minimizer of equation 4, we have

$$\left\|\mathcal{M}\odot(\boldsymbol{M}-\boldsymbol{Z}^{(i+1)})\right\|_F^2 + \rho\left\|\boldsymbol{Z}^{(i+1)}-\boldsymbol{R}^{(i+1)}+\frac{1}{\rho}\alpha_{\boldsymbol{Z}}^{(i)}\right\|_F^2$$

$$\leq\left\|\mathcal{M}\odot(\boldsymbol{M}-\boldsymbol{Z}^{(i)})\right\|_F^2 + \rho\left\|\boldsymbol{Z}^{(i)}-\boldsymbol{R}^{(i+1)}+\frac{1}{\rho}\alpha_{\boldsymbol{Z}}^{(i)}\right\|_F^2$$

which gives

$$2\left\langle\mathcal{M}\odot(\boldsymbol{Z}^{(i+1)}-\boldsymbol{M}),\mathcal{M}\odot(\boldsymbol{Z}^{(i+1)}-\boldsymbol{Z}^{(i)})\right\rangle - \|\mathcal{M}\odot(\boldsymbol{Z}^{(i+1)}-\boldsymbol{Z}^{(i)})\|_F^2$$

$$\leq -2\left\langle\rho\cdot\boldsymbol{Z}^{(i+1)}+\alpha_{\boldsymbol{Z}}^{(i)}-\rho\boldsymbol{R}^{(i+1)},\boldsymbol{Z}^{(i+1)}-\boldsymbol{Z}^{(i)}\right\rangle + \rho\|\boldsymbol{Z}^{(i+1)}-\boldsymbol{Z}^{(i)}\|_F^2$$

It further implies

$$\left\langle\rho\cdot\boldsymbol{Z}^{(i+1)}+\alpha_{\boldsymbol{Z}}^{(i)}-\rho\boldsymbol{R}^{(i+1)},\boldsymbol{Z}^{(i+1)}-\boldsymbol{Z}^{(i)}\right\rangle$$

$$\leq -\left\langle\mathcal{M}\odot(\boldsymbol{Z}^{(i+1)}-\boldsymbol{M}),\mathcal{M}\odot(\boldsymbol{Z}^{(i+1)}-\boldsymbol{Z}^{(i)})\right\rangle + \frac{1}{2}\|\mathcal{M}\odot(\boldsymbol{Z}^{(i+1)}-\boldsymbol{Z}^{(i)})\|_F^2$$

$$+\frac{\rho}{2}\|\boldsymbol{Z}^{(i+1)}-\boldsymbol{Z}^{(i)}\|_F^2$$

By direct substitution, we have

$$(\mathcal{A})\leq\left\langle\mathcal{M}\odot(\boldsymbol{Z}^{(i+1)}-\boldsymbol{M}),\boldsymbol{Z}^{(i+1)}-\boldsymbol{Z}^{(i)}\right\rangle$$

$$-\left\langle\mathcal{M}\odot(\boldsymbol{Z}^{(i+1)}-\boldsymbol{M}),\mathcal{M}\odot(\boldsymbol{Z}^{(i+1)}-\boldsymbol{Z}^{(i)})\right\rangle$$

$$+\frac{1}{2}\|\mathcal{M}\odot(\boldsymbol{Z}^{(i+1)}-\boldsymbol{Z}^{(i)}\|_F^2 + \frac{\rho}{2}\|\boldsymbol{Z}^{(i+1)}-\boldsymbol{Z}^{(i)}\|_F^2 - \|\mathcal{M}\odot(\boldsymbol{Z}^{(i+1)}-\boldsymbol{Z}^{(i)})\|_F^2$$

$$-\rho\|(\boldsymbol{Z}^{(i+1)}-\boldsymbol{Z}^{(i)})\|_F^2 - \left\langle\mathcal{M}\odot(\boldsymbol{Z}^{(i+1)}-\boldsymbol{M}),(1-\mathcal{M})\odot(\boldsymbol{Z}^{(i+1)}-\boldsymbol{Z}^{(i)})\right\rangle$$

$$= -\frac{1}{2}\|\mathcal{M}\odot(\boldsymbol{Z}^{(i+1)}-\boldsymbol{Z}^{(i)})\|_F^2 - \frac{\rho}{2}\|(\boldsymbol{Z}^{(i+1)}-\boldsymbol{Z}^{(i)})\|_F^2 \leq -\frac{\rho}{2}\|(\boldsymbol{Z}^{(i+1)}-\boldsymbol{Z}^{(i)})\|_F^2 \quad (19)$$

For the second term $(\mathcal{B})$, by definition, we have,

$$(\mathcal{B}) = \frac{\rho}{2}\left\|\boldsymbol{Z}^{(i)}-\boldsymbol{R}^{(i+1)}+\frac{1}{\rho}\alpha_{\boldsymbol{Z}}^{(i)}\right\|_F^2 - \frac{\rho}{2}\left\|\boldsymbol{Z}^{(i)}-[\boldsymbol{W}^{(i+1)},\boldsymbol{C}]\boldsymbol{Q}^{(i),T}+\frac{1}{\rho}\alpha_{\boldsymbol{Z}}^{(i)}\right\|_F^2$$

$$+\beta\|\boldsymbol{Q}^{(i+1)}\|_{1,1} - \beta\|\boldsymbol{Q}^{(i)}\|_{1,1}$$

$$= \rho\left\langle\boldsymbol{R}^{(i+1)}-\boldsymbol{Z}^{(i)}-\frac{1}{\rho}\alpha_{\boldsymbol{Z}}^{(i)},[\boldsymbol{W}^{(i+1)},\boldsymbol{C}](\boldsymbol{Q}^{(i+1),T}-\boldsymbol{Q}^{(i),T})\right\rangle$$

$$-\frac{\rho}{2}\left\|[\boldsymbol{W}^{(i+1)},\boldsymbol{C}](\boldsymbol{Q}^{(i+1),T}-\boldsymbol{Q}^{(i),T})\right\|_F^2 + \beta(\|\boldsymbol{Q}^{(i+1)}\|_{1,1}-\|\boldsymbol{Q}^{(i)}\|_{1,1})$$

We recall that $\boldsymbol{Q}$ is updated via solving constrained Lasso problems for every row $\boldsymbol{Q}_{[r,:]}^{(i+1)}$:

$$\boldsymbol{y} = \underset{\boldsymbol{x},0\leq\boldsymbol{x}}{\arg\min}\ \beta\|\boldsymbol{x}\|_1 + \frac{\rho}{2}\|\boldsymbol{b}-\boldsymbol{Ax}\|_2^2, \quad \text{where}\quad \boldsymbol{A}=[\boldsymbol{W}^{(i+1)},\boldsymbol{C}], \boldsymbol{b}=\left[\boldsymbol{Z}^{(i)}+\frac{1}{\rho}\alpha_{\boldsymbol{Z}}^{(i)}\right]_{[r,:]} \quad (20)$$

One obtains $\boldsymbol{y}$ if and only if there exists $\boldsymbol{g}\in\partial\|\boldsymbol{y}\|_1$, the sub-differential of $\|\cdot\|_1$ such that

$$\rho\boldsymbol{A}^T(\boldsymbol{Ay}-\boldsymbol{b})+\beta\boldsymbol{g}=\boldsymbol{0}. \quad (21)$$

As $\|\cdot\|_1$ is convex, we have

$$\|\boldsymbol{x}\|_1\geq\|\boldsymbol{y}\|_1+\langle\boldsymbol{x}-\boldsymbol{y},\boldsymbol{g}\rangle \quad (22)$$

which gives

$$\|\boldsymbol{y}\|_1-\|\boldsymbol{x}\|_1\leq\left\langle\boldsymbol{y}-\boldsymbol{x},\frac{\rho}{\beta}\boldsymbol{A}^T(\boldsymbol{Ay}-\boldsymbol{b})\right\rangle=\left\langle\boldsymbol{A}(\boldsymbol{y}-\boldsymbol{x}),\frac{\rho}{\beta}(\boldsymbol{Ay}-\boldsymbol{b})\right\rangle \quad (23)$$

Re-substituting $\boldsymbol{x} = \boldsymbol{Q}_{[r,:]}^{(i),T}$, $\boldsymbol{y} = \boldsymbol{Q}_{[r,:]}^{(i+1),T}$, $\boldsymbol{A} = [\boldsymbol{W}^{(i+1)}, \boldsymbol{C}]$, $\boldsymbol{b} = \left[\boldsymbol{Z}^{(i)} + \frac{1}{\rho}\alpha_{\boldsymbol{Z}}^{(i)}\right]_{[r,:]}$ and sum over $r$, we have

$$\beta\|\boldsymbol{Q}^{(i+1)}\|_{1,1} - \beta\|\boldsymbol{Q}^{(i)}\|_{1,1} \leq -\rho\left\langle \boldsymbol{R}^{(i+1)} - \boldsymbol{Z}^{(i)} - \frac{1}{\rho}\alpha_{\boldsymbol{Z}}^{(i)}, [\boldsymbol{W}^{(i+1)}, \boldsymbol{C}](\boldsymbol{Q}^{(i+1),T} - \boldsymbol{Q}^{(i),T})\right\rangle \tag{24}$$

Therefore, we have

$$(\mathcal{B}) \leq -\frac{\rho}{2}\left\|[\boldsymbol{W}^{(i+1)}, \boldsymbol{C}](\boldsymbol{Q}^{(i+1),T} - \boldsymbol{Q}^{(i),T})\right\|_F^2 \tag{25}$$

With similar argument, we can bound $(\mathcal{C})$ by

$$(\mathcal{C}) \leq -\frac{\rho}{2}\left\|[(\boldsymbol{W}^{(i+1)} - \boldsymbol{W}^{(i)}), \boldsymbol{C}]\boldsymbol{Q}^{(i),T}\right\|_F^2 \tag{26}$$

To get an upper bound of $\|\alpha_{\boldsymbol{Z}}^{(i+1)} - \alpha_{\boldsymbol{Z}}^{(i)}\|_F^2$, we have

$$\|\alpha_{\boldsymbol{Z}}^{(i+1)} - \alpha_{\boldsymbol{Z}}^{(i)}\|_F^2$$
$$\leq \|\boldsymbol{Z}^{(i+1)} - \boldsymbol{Z}^{(i)}\|_F^2 + \|\boldsymbol{R}^{(i+1)} - \boldsymbol{R}^{(i)}\|_F^2$$
$$\leq \|\boldsymbol{Z}^{(i+1)} - \boldsymbol{Z}^{(i)}\|_F^2 + \|[\boldsymbol{W}^{(i+1)}, \boldsymbol{C}]\boldsymbol{Q}^{(i+1),T} - [\boldsymbol{W}^{(i+1)}, \boldsymbol{C}]\boldsymbol{Q}^{(i),T}\|_F^2$$
$$\quad + \|[\boldsymbol{W}^{(i+1)}, \boldsymbol{C}]\boldsymbol{Q}^{(i),T} - [\boldsymbol{W}^{(i)}, \boldsymbol{C}]\boldsymbol{Q}^{(i),T}\|_F^2$$
$$\leq \|\boldsymbol{Z}^{(i+1)} - \boldsymbol{Z}^{(i)}\|_F^2 + \|[\boldsymbol{W}^{(i+1)}, \boldsymbol{C}](\boldsymbol{Q}^{(i+1),T} - \boldsymbol{Q}^{(i),T})\|_F^2 + \|[(\boldsymbol{W}^{(i+1)} - \boldsymbol{W}^{(i)}), \boldsymbol{C}]\boldsymbol{Q}^{(i),T}\|_F^2 \tag{27}$$

Combining equation 17, 27, 18, 19, 25 and 26 with equation 16, we have

$$\mathcal{L}_\rho(\mathbb{V}^{(i+1)}, \alpha_{\boldsymbol{Z}}^{(i+1)}) - \mathcal{L}_\rho(\mathbb{V}^{(i)}, \alpha_{\boldsymbol{Z}}^{(i)})$$
$$\leq \frac{1}{\rho}\left\|\alpha_{\boldsymbol{Z}}^{(i+1)} - \alpha_{\boldsymbol{Z}}^{(i)}\right\|_F^2 - \frac{\rho}{2}\left\|\boldsymbol{Z}^{(i+1)} - \boldsymbol{Z}^{(i)}\right\|_F^2 - \frac{\rho}{2}\left\|[\boldsymbol{W}^{(i+1)}, \boldsymbol{C}](\boldsymbol{Q}^{(i+1),T} - \boldsymbol{Q}^{(i),T})\right\|_F^2$$
$$\quad - \frac{\rho}{2}\left\|[(\boldsymbol{W}^{(i+1)} - \boldsymbol{W}^{(i)}), \boldsymbol{C}]\boldsymbol{Q}^{(i),T}\right\|_F^2$$
$$\leq \left(\frac{1}{\rho} - \frac{\rho}{2}\right)\cdot\left(\|\boldsymbol{Z}^{(i+1)} - \boldsymbol{Z}^{(i)}\|_F^2 + \|[\boldsymbol{W}^{(i+1)}, \boldsymbol{C}](\boldsymbol{Q}^{(i+1),T} - \boldsymbol{Q}^{(i),T})\|_F^2\right.$$
$$\left. + \|[(\boldsymbol{W}^{(i+1)} - \boldsymbol{W}^{(i)}), \boldsymbol{C}]\boldsymbol{Q}^{(i),T}\|_F^2\right) \tag{28}$$

This is summarized into the following theorem:

*Theorem* B.1 (Non-increasing property). Assume $\rho \geq \sqrt{2}$, for all $i$, we have

$$\mathcal{L}_\rho(\boldsymbol{W}^{(i+1)}, \boldsymbol{Q}^{(i+1)}, \boldsymbol{Z}^{(i+1)}, \alpha_{\boldsymbol{Z}}^{(i+1)}) \leq \mathcal{L}_\rho(\boldsymbol{W}^{(i)}, \boldsymbol{Q}^{(i)}, \boldsymbol{Z}^{(i)}, \alpha_{\boldsymbol{Z}}^{(i)}). \tag{29}$$

We set $\rho = 3$ in all experiments for sufficiency.

## C   ERROR ANALYSIS WHEN $k \neq k^*$

Assume that there is a ground-truth factorization $(\mathbf{W}^*, \mathbf{Q}^*)$ of the given $\mathbf{M} = \mathbf{W}^*(\mathbf{Q}^*)^T$, with latent dimension $k^*$, where $\mathbf{W}^*$ and $\mathbf{Q}^*$ are matrix-valued random variables with entries sampled from some bounded distributions. With high probability, the error $\|\mathbf{M} - \mathbf{W}\mathbf{Q}^T\|_F^2$ we are minimizing is star-convex towards $(\mathbf{W}^*, \mathbf{Q}^*)$ whenever $k = k^*$ (Bjorck et al., 2021). To demonstrate the importance of the choice of $k$, we consider the scenario when $k \neq k^*$ below.

First, a more precise assumption for ICQF is to model $\mathbf{W}$ as *row-independent bounded random matrices*. Recall that $\boldsymbol{W}$ is generated by arranging $n$ participants' latent representation as rows of $n \times k$ matrix, where the $n$ participants are assumed to be independent from each other and their corresponding latent representations follow a high-dimensional bounded distribution.

Second, let $(\mathbf{W}_1, \mathbf{Q}_1)$ and $(\mathbf{W}_2, \mathbf{Q}_2)$ be two factorizations with dimensions $k_1$ and $k_2$ respectively. Assume both factorizations achieve **(a)**: equivalent mismatching loss in expectation, and **(b)**: equivalent expectation approximation to data matrix $\mathbf{M}$:

$$\textbf{(a)}: \; \mathbb{E}\left[\|\mathbf{M} - \mathbf{W}_1\mathbf{Q}_1^T\|_F^2\right] = \mathbb{E}\left[\|\mathbf{M} - \mathbf{W}_2\mathbf{Q}_2^T\|_F^2\right] \quad \text{and} \quad \textbf{(b)}: \; \mathbb{E}[\mathbf{W}_1\mathbf{Q}_1^T] = \mathbb{E}[\mathbf{W}_2\mathbf{Q}_2^T]$$

We also assume **(c)**: $\mathbb{E}\left[\sum_{j=1}^n (\mathbf{W}_i)_{j\kappa}^2\right] := \sigma_{\mathbf{W}_i}^2$ and $\mathbb{E}\left[\sum_{j=1}^m (\mathbf{Q}_i)_{j\kappa}^2\right] := \sigma_{\mathbf{Q}_i}^2$ for all $\kappa = k_i$, $i = 1, 2$.

Expanding **(a)**, we have

$$\mathbb{E}\left[\text{Trace}\left((\mathbf{M} - \mathbf{W}_1\mathbf{Q}_1^T)^T(\mathbf{M} - \mathbf{W}_1\mathbf{Q}_1^T)\right)\right] = \mathbb{E}\left[\text{Trace}\left((\mathbf{M} - \mathbf{W}_2\mathbf{Q}_2^T)^T(\mathbf{M} - \mathbf{W}_2\mathbf{Q}_2^T)\right)\right]$$

This gives

$$\mathbb{E}\left[\text{Trace}\left(\mathbf{W}_1^T\mathbf{W}_1\mathbf{Q}_1^T\mathbf{Q}_1 - 2\mathbf{M}^T\mathbf{W}_1\mathbf{Q}_1^T\right)\right] = \mathbb{E}\left[\text{Trace}\left(\mathbf{W}_2^T\mathbf{W}_2\mathbf{Q}_2^T\mathbf{Q}_2 - 2\mathbf{M}^T\mathbf{W}_2\mathbf{Q}_2^T\right)\right]$$

Denote $\mathbb{E}[\mathbf{W}_i] = \mu_{\mathbf{W}_i}$, $\mathbb{E}[\mathbf{Q}_i] = \mu_{\mathbf{Q}_i}$ for $i = 1, 2$, we have $\mathbf{W}_i = \bar{\mathbf{W}}_i + \mu_{\mathbf{W}_i}$ and $\mathbf{Q}_i = \bar{\mathbf{Q}}_i + \mu_{\mathbf{Q}_i}$, where $\bar{\mathbf{W}}_i$ and $\bar{\mathbf{Q}}_i$ denote the corresponding centered variables. Note that by the independence of $\mathbf{W}_i$ and $\mathbf{Q}_i$ and linearity of trace and expectation operator,

$$\begin{aligned} &\mathbb{E}\left[\text{Trace}\left(\mathbf{M}^T\mathbf{W}_1\mathbf{Q}_1^T\right)\right]\\ =&\mathbb{E}\left[\text{Trace}\left(\mathbf{M}^T\bar{\mathbf{W}}_1\bar{\mathbf{Q}}_1^T + \mathbf{M}^T\bar{\mathbf{W}}_1\mu_{\mathbf{Q}_1}^T + \mathbf{M}^T\mu_{\mathbf{W}_1}\bar{\mathbf{Q}}_1^T + \mathbf{M}^T\mu_{\mathbf{W}_1}\mu_{\mathbf{Q}_1}^T\right)\right]\\ =&\text{Trace}(\mathbf{M}^T\mathbb{E}[\mathbf{W}_1]\mathbb{E}[\mathbf{Q}_1^T]) = \text{Trace}(\mathbf{M}^T\mathbb{E}[\mathbf{W}_2]\mathbb{E}[\mathbf{Q}_2^T]) = \mathbb{E}\left[\text{Trace}\left(\mathbf{M}^T\mathbf{W}_2\mathbf{Q}_2^T\right)\right] \end{aligned} \tag{30}$$

which yields

$$\mathbb{E}\left[\text{Trace}\left(\mathbf{W}_1^T\mathbf{W}_1\mathbf{Q}_1^T\mathbf{Q}_1\right)\right] = \mathbb{E}\left[\text{Trace}\left(\mathbf{W}_2^T\mathbf{W}_2\mathbf{Q}_2^T\mathbf{Q}_2\right)\right] \tag{31}$$

Consider $\mathbb{E}\left[\text{Trace}\left(\mathbf{W}_1^T\mathbf{W}_1\mathbf{Q}_1^T\mathbf{Q}_1\right)\right]$ via definition, we have

$$\begin{aligned} &\mathbb{E}\left[\text{Trace}\left(\mathbf{W}_1^T\mathbf{W}_1\mathbf{Q}_1^T\mathbf{Q}_1\right)\right]\\ =&\text{Trace}\left(\mathbb{E}\left[\mathbf{W}_1^T\mathbf{W}_1\right]\mathbb{E}\left[\mathbf{Q}_1^T\mathbf{Q}_1\right]\right)\\ =&\text{Trace}\left(\mathbb{E}\begin{bmatrix} \left(\sum_{j=1}^n(\mathbf{W}_1)_{j1}^2\right) & & * \\ & \ddots & \\ * & & \left(\sum_{j=1}^n(\mathbf{W}_1)_{jk_1}^2\right) \end{bmatrix}\right.\\ &\left.\times \mathbb{E}\begin{bmatrix} \left(\sum_{j=1}^m(\mathbf{Q}_1)_{j1}^2\right) & & 0 \\ & \ddots & \\ 0 & & \left(\sum_{j=1}^m(\mathbf{Q}_1)_{jk_1}^2\right) \end{bmatrix}\right)\\ =&\sum_{\kappa=1}^{k_1}\mathbb{E}\left[\sum_{j=1}^n(\mathbf{W}_1)_{j\kappa}^2\right]\mathbb{E}\left[\sum_{j=1}^m(\mathbf{Q}_1)_{j\kappa}^2\right] \end{aligned} \tag{32}$$

Incorporating assumption **(c)**, we have

$$\mathbb{E}\left[\text{Trace}\left(\mathbf{W}_1^T\mathbf{W}_1\mathbf{Q}_1^T\mathbf{Q}_1\right)\right] = k_1\sigma_{\mathbf{W}_1}^2\sigma_{\mathbf{Q}_1}^2 \tag{33}$$

Consider equation 31 with $k_1 > k_2$. For $\mathbf{W}_1, \mathbf{Q}_1$, W.L.O.G. we pad $k_2 - k_1$ columns of zeros. Moreover, let $\mathbf{P}$ be an optimal $k_2 \times k_2$ permutation matrix, we also have

$$\mathbb{E}\left[\text{Trace}\left((\mathbf{W}_2\mathbf{P})^T\mathbf{W}_2\mathbf{P}(\mathbf{Q}_2\mathbf{P})^T\mathbf{Q}_2\mathbf{P}\right)\right] = \mathbb{E}\left[\text{Trace}\left(\mathbf{W}_2^T\mathbf{W}_2\mathbf{Q}_2^T\mathbf{Q}_2\right)\right] = k_2\sigma_{\mathbf{W}_2}^2\sigma_{\mathbf{Q}_2}^2 \tag{34}$$

Combining with equation 31, it is equivalent to

$$k_1\sigma_{\mathbf{W}_1}^2\sigma_{\mathbf{Q}_1}^2 = k_2\sigma_{\mathbf{W}_2}^2\sigma_{\mathbf{Q}_2}^2 \tag{35}$$

which gives

$$\mathbb{E}\left[\|\mathbf{W}_1\|_F^2\right] = \frac{\sigma_{\mathbf{Q}_2}^2}{\sigma_{\mathbf{Q}_1}^2}\mathbb{E}\left[\|\mathbf{W}_2\|_F^2\right] = \frac{\sigma_{\mathbf{Q}_2}^2}{\sigma_{\mathbf{Q}_1}^2}\mathbb{E}\left[\|\mathbf{W}_2\mathbf{P}\|_F^2\right] \tag{36}$$

To evaluate the impact of interpretability of latent representation under different latent dimension, we consider $\mathbb{E}\left[\|\mathbf{W}_1 - \mathbf{W}_2\mathbf{P}\|_F^2\right]$:

$$\mathbb{E}\left[\|\mathbf{W}_1 - \mathbf{W}_2\mathbf{P}\|_F^2\right] = \mathbb{E}\left[\text{Trace}\left((\mathbf{W}_1 - \mathbf{W}_2\mathbf{P})^T(\mathbf{W}_1 - \mathbf{W}_2\mathbf{P})\right)\right]$$

$$= \mathbb{E}\left[\|\mathbf{W}_1\|_F^2\right] + \frac{\sigma_{\mathbf{Q}_1}^2}{\sigma_{\mathbf{Q}_2}^2}\mathbb{E}\left[\|\mathbf{W}_1\|_F^2\right] - 2\mathbb{E}\left[\text{Trace}(\mathbf{W}_1^T\mathbf{W}_2\mathbf{P})\right] \quad (37)$$

As $\text{Trace}(\mathbf{W}_1^T\mathbf{W}_2 P) \leq \|\mathbf{W}_1\|_F\|\mathbf{W}_2\mathbf{P}\|_F$, we also have

$$\mathbb{E}\left[\text{Trace}(\mathbf{W}_1^T\mathbf{W}_2\mathbf{P})\right] \leq \mathbb{E}\left[\|\mathbf{W}_1\|_F\right] \cdot \mathbb{E}\left[\|\mathbf{W}_2\mathbf{P}\|_F\right]$$

$$\leq \sqrt{\mathbb{E}\left[\|\mathbf{W}_1\|_F^2\right] \cdot \mathbb{E}\left[\|\mathbf{W}_2\|_F^2\right]} = \sqrt{\frac{\sigma_{\mathbf{Q}_1}^2}{\sigma_{\mathbf{Q}_2}^2}}\mathbb{E}\left[\|\mathbf{W}_1\|_F^2\right] \quad (38)$$

which implies

$$\mathbb{E}\left[\|\mathbf{W}_1 - \mathbf{W}_2\mathbf{P}\|_F^2\right] \geq \left(1 - 2\sqrt{\frac{\sigma_{\mathbf{Q}_1}^2}{\sigma_{\mathbf{Q}_2}^2}} + \frac{\sigma_{\mathbf{Q}_1}^2}{\sigma_{\mathbf{Q}_2}^2}\right)\mathbb{E}\left[\|\mathbf{W}_1\|_F^2\right] = \left(1 - \sqrt{\frac{\sigma_{\mathbf{Q}_1}^2}{\sigma_{\mathbf{Q}_2}^2}}\right)^2\mathbb{E}\left[\|\mathbf{W}_1\|_F^2\right]$$
$$(39)$$

Since $\mathbf{W}_i$ is generated from row-wise independent bounded distribution, if we add a mild assumption that $\sigma_{\mathbf{W}_i}^2 := \sigma_{\mathbf{W}}^2$ for all $i$ through re-scaling, Equation 35 implies $k_1\sigma_{\mathbf{Q}_1}^2 = k_2\sigma_{\mathbf{Q}_2}^2$ and therefore

$$\mathbb{E}\left[\|\mathbf{W}_1 - \mathbf{W}_2\|_F^2\right] \geq \left(1 - 2\sqrt{\frac{k_2}{k_1}} + \frac{k_2}{k_1}\right)\mathbb{E}\left[\|\mathbf{W}_1\|_F^2\right] = \left(\sqrt{\frac{k_2}{k_1}} - 1\right)^2\mathbb{E}\left[\|\mathbf{W}_1\|_F^2\right] \quad (40)$$

If we substitute $k_1 = k^*$, $(\mathbf{W}_1, \mathbf{Q}_1) = (\mathbf{W}^*, \mathbf{Q}^*)$, we have

$$\mathbb{E}\left[\|\mathbf{W}^* - \mathbf{W}_2\|_F^2\right] \geq \left(\sqrt{\frac{k_2}{k^*}} - 1\right)^2\mathbb{E}\left[\|\mathbf{W}^*\|_F^2\right] \quad (41)$$

which means the relative expected difference between $\mathbf{W}^*$ and $\mathbf{W}_2$ is bounded below by $\left(\sqrt{\frac{k_2}{k^*}} - 1\right)^2$.

To prove that equation 41 holds in general, we consider the matrix concentration inequalities and show that large deviations from their means are exponentially unlikely. Benefitting from the model constraints, we can further assume that $W$ is generated from some high dimensional bounded distribution. In the following, we make use of the main theorem proposed in Meckes & Szarek (2012) on concentration of non-commutative random matrices polynomials. As $\mathbf{W}_i$ are generated from bounded distributions, $\|\mathbf{W}_i - \mathbb{E}[\mathbf{W}_i]\|_F$ is uniformly bounded. Therefore, it satisfies the convex concentration properties. The theorem achieves the following results:

$$\mathbb{P}\left\{\|\mathbf{W}\|_F^2 - \mathbb{E}\left[\|\mathbf{W}\|_F^2\right] > tkn^2\right\} \leq C_1\exp\left(-C_2\min(t^2, t^{1/2})n\right) \quad (42)$$

Recall that $\mathbb{E}\left[\|\mathbf{W}_1 - \mathbf{W}_2 P\|_F^2\right] = \mathbb{E}\left[\|\mathbf{W}_1\|_F^2\right] + \frac{\sigma_{\mathbf{Q}_1}^2}{\sigma_{\mathbf{Q}_2}^2}\mathbb{E}\left[\|\mathbf{W}_1\|_F^2\right] - 2\mathbb{E}\left[\text{Trace}(\mathbf{W}_1^T\mathbf{W}_2\mathbf{P})\right]$. By padding $\mathbf{W}_1$ and $\mathbf{W}_2$ with zeros columns, we assume that $\mathbf{W}_i$ are all $n \times n$ matrices. Then the probability that the any one of the terms is deviating from their mean by a relative factor $\epsilon$ is less than $C_1\exp(-C_2\epsilon^2 n)$ for some small $\epsilon$. By the union bound, the probability that the either of them does is less than or equal to $C_3\exp(-C_4\epsilon^2 n)$.

# D VISUALIZATION OF THE EXPERIMENTAL SETUP FOR DIAGNOSTIC PREDICTION EVALUATION

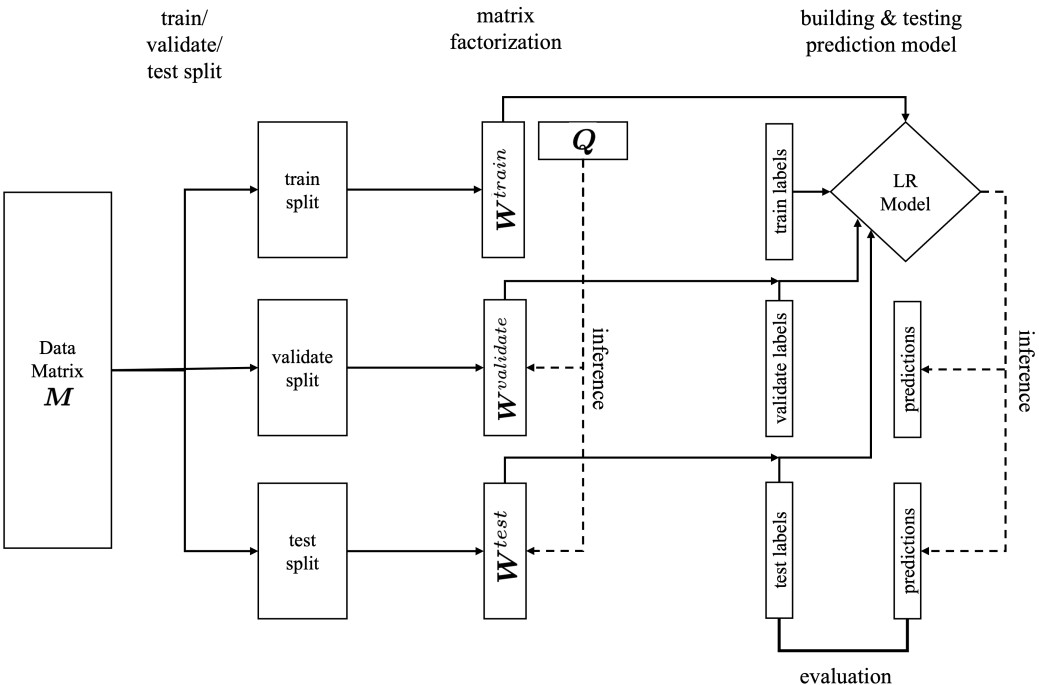

Figure 4: Setup for diagnostic prediction experiments.

# E   VISUALIZATION OF TRAIN, VALIDATION AND TEST SETS IN MULTIPLE QUESTIONNAIRES SETTING

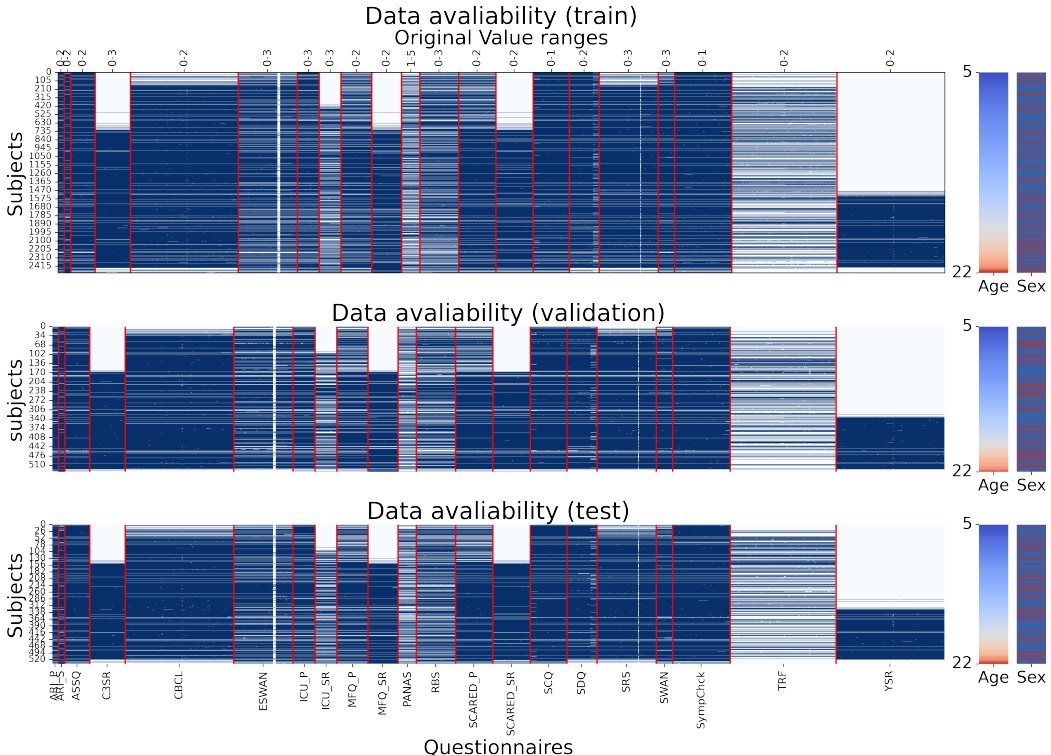

Figure 5: Train, validation and test split of one data re-sampling.

Table 2: Number of positive diagnostic labels in *HBN* dataset

| Diagnosis | number of positive cases |
|---|---|
| Depression | 211 |
| GenAnxiety | 262 |
| ADHD | 506 |
| Suspected ASD | 222 |
| Panic, Agoraphobia, Separation Anxiety, Social Anxiety | 327 |
| BPD | 299 |
| Specific phobia | 234 |
| OCD | 64 |
| Eating Disorder | 97 |
| PTSD Trauma | 27 |
| Sleep problems | 118 |

# F   DETAILS OF THE SYNTHETIC QUESTIONNAIRE EXPERIMENT

This experiment aimed at comparing the effectiveness of BCV and other procedures for estimating the number of latent factors in a synthetic example, for ICQF and two baseline methods. We generated the synthetic questionnaire with $k^* = 10$ factors ($[0, 1]$ bounded), where each factor was present in isolation for 200 participants and in tandem with another factor for others. Each factor had an associated loading over 100 questions. The answers to questions are bounded between 0 and 100.

To generate the answer matrix $\boldsymbol{M} \in \mathbb{R}_{\geq 0}^{200 \times 100}$, we design the underlying matrices $\boldsymbol{W}$ and $\boldsymbol{Q}$ directly. We first create a matrix $\boldsymbol{D}$ of size $200 \times 10$, shown in Figure 6 (left). The 10 columns of $\boldsymbol{D}$ are correlated with a step-like pattern, where each "step" is of length 20 and entries on the "step" have weight 1. Every consecutive pair of steps is overlapped by 10 units to synthesize correlation between latent factors. An entry $\boldsymbol{W}[i, j]$ is then defined to be

$$\boldsymbol{W}[i, j] := \boldsymbol{D}[i, j] * a * b, \quad a \sim U(0.5, 1), \; b \sim B(1, p = 0.9) \tag{43}$$

where $U(0.5, 1)$ is the uniform distribution between $[0.5, 1]$ and $B(1, p = 0.9)$ is the Bernoulli distribution with probability $p = 0.9$. The matrix $\boldsymbol{Q}$ of size $100 \times 10$, shown in Figure 6 (center) is defined to be

$$\boldsymbol{Q}[i, j] := c * d, \quad c \sim U(0, 100), \; d \sim B(1, 0.3) \tag{44}$$

Having $\boldsymbol{W}$ and $\boldsymbol{Q}$, we obtain a noiseless data matrix $\boldsymbol{M}_{clean} := \min(0, \max(\boldsymbol{W}\boldsymbol{Q}^T, 100))$. To introduce additive noise, we modify $\boldsymbol{M}_{clean}$ by

$$\boldsymbol{M} := \min\left(0, \max(\boldsymbol{M}_{clean} + e * f, 100)\right), \quad f \sim U(-100, 100) \tag{45}$$

where $e$ follows a discrete probability distribution with $P(e = 1) = \delta, P(e = 0) = 1 - \delta$. This yields a data matrix $\boldsymbol{M}$, shown in Figure 6 (right) for $\delta = 0.3$. If $\delta$ is high, more entries in the $\boldsymbol{M}_{clean}$ will be contaminated by the additive noise $f$.

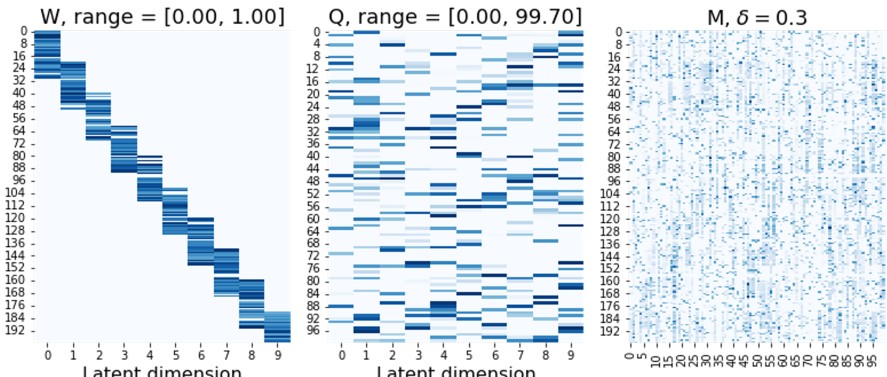

Figure 6: Synthetic example of $\boldsymbol{W}$, $\boldsymbol{Q}$ and $\boldsymbol{M}$ with noise density $\delta = 0.3$.

We compared ICQF against $\ell_1$-regularized NMF ($\ell_1$-NMF) (Cichocki & Phan, 2009) and factor analysis with promax rotation (FA-promax) (Hendrickson & White, 1964), as factors can be correlated. Both ICQF and $\ell_1$-NMF were initialized with NNDSVD (Boutsidis & Gallopoulos, 2008), and the sparsity ($\beta = 1e-1$) and stopping criterion (relative iteration convergence tolerance $\epsilon < 1e-3$) for fairness. The estimation method for FA was minimum residual.

Table 3 shows the mean error $\bar{\epsilon}$ and the standard error $s_E$ of the detected $k$ versus ground-truth $k^* = 10$, across different generated datasets. Statistics are marked with asterisk if they contain runs where no inflection point was detected using the kneed algorithm (Satopaa et al., 2011). We tested five popular detection algorithms: BCV (Kanagal & Sindhwani, 2010), $\{BIC_1, BIC_2\}$ (Stoica & Selen, 2004)[3], CCC (Fogel et al., 2007) and Dispersion (Brunet et al., 2004). For ICQF, BCV is the best overall detection scheme at all noise levels; $BIC_2$ performs well for low noise only. Within the three common schemes for FA, Horn's PA (Horn, 1965) and MAP (Velicer, 1976) are superior to $BIC_3$ (Preacher et al., 2013). PA achieves the best performance when the noise density is high. This result aligns with the conclusions in Velicer et al. (2000); Watkins (2018); Goretzko et al. (2021).

---

[3]The two $BIC$ versions are respectively $BIC_1(k) := \log\left(\|\boldsymbol{M} - \boldsymbol{W}\boldsymbol{Q}^T\|_F^2\right) + k\frac{m+n}{mn}\log\left(\frac{mn}{m+n}\right)$ and $BIC_2(k) := \log\left(\|\boldsymbol{M} - \boldsymbol{W}\boldsymbol{Q}^T\|_F^2\right) + k\frac{m+n}{mn}\log\left(\min(\sqrt{m}, \sqrt{n})^2\right)$. There are other similar versions of $BIC$ but their performances are indistinguishable, thus we only include two representatives in the manuscript.

Table 3: Experiment of $k$ estimation for synthetic questionnaires. The average error ($\bar{\epsilon}$) and the standard error of estimated $k$ ($s_E$) are reported. Statistics with asterisk (*) ignore undetectable inflection points.

| Methods | Noise density $\delta$ | Detection schemes for NMF | | | | |
|---|---|---|---|---|---|---|
| | | BCV | $BIC_1$ | $BIC_2$ | $CCC$ | Dispersion |
| ICQF | 0.1 | $(\mathbf{0.10}, \mathbf{0.06})$ | $(0.13, 0.08)$ | $(\mathbf{0.10}, \mathbf{0.06})$ | $(1.33, 0.24)$ | $(0.23, 0.09)$ |
| | 0.2 | $(\mathbf{0.11}, \mathbf{0.06})$ | $(1.23, 0.34)^*$ | $(0.67, 0.23)^*$ | $(1.14, 0.21)$ | $(0.93, 0.16)$ |
| | 0.3 | $(\mathbf{0.77}, \mathbf{0.15})$ | NaN$^*$ | $(2.40, 0.48)^*$ | $(0.96, 0.18)$ | $(2.60, 0.26)$ |
| $\ell_1$-NMF | 0.1 | $(0.17, 0.07)$ | $(1.20, 0.21)^*$ | $(0.90, 0.23)^*$ | NaN$^*$ | NaN$^*$ |
| | 0.2 | $(2.37, 0.33)$ | NaN$^*$ | $(1.10, 0.30)^*$ | NaN$^*$ | NaN$^*$ |
| | 0.3 | $(2.40, 0.31)$ | NaN$^*$ | $(2.47, 0.54)^*$ | NaN$^*$ | NaN$^*$ |
| | | Detection schemes for FA | | | | |
| | | PA | MAP | $BIC_3$ | | |
| FA-promax | 0.1 | $(0.17, 0.07)$ | $(\mathbf{0.11}, \mathbf{0.06})$ | $(0.30, 0.03)$ | | |
| | 0.2 | $(0.53, 0.10)$ | $(\mathbf{0.13}, \mathbf{0.06})$ | $(0.93, 0.11)$ | | |
| | 0.3 | $(\mathbf{0.87}, \mathbf{0.14})$ | $(1.27, 0.20)$ | NaN$^*$ | | |

# G  TABLE SUMMARIZING THE OPTIMAL $(k, \beta)$ OF THE 21 QUESTIONNAIRES

Table 4: Optimal $(k, \beta)$ of all 21 questionnaires.

| Questionnaire | Abbreviation | $n$ questions | Subscales | $k$ | $\beta$ |
|---|---|---|---|---|---|
| Affective Reactivity Index (Parent-Report) | ARI_P | 7 | nan | 2 | 0.01 |
| Affective Reactivity Index (Self-Report) | ARI_S | 7 | nan | 2 | 0.01 |
| Autism Spectrum Screening Questionnaire | ASSQ | 27 | nan | 2 | 0.01 |
| Conners 3 (Self-Report) | C3SR | 9 | | 4 | 0.05 |
| Child Behavior Checklist | CBCL | 119 | 9 | 8 | 0.5 |
| Extended Strengths and Weaknesses Assessment of Normal Behavior | ESWAN | 65 | nan | 13 | 0.2 |
| Inventory of Callous-Unemotional Traits (Parent-Report) | ICU_P | 24 | 3 | 4 | 0.1 |
| Inventory of Callous-Unemotional Traits (Self-Report) | ICU_SR | 24 | 3 | 3 | 0.1 |
| Mood and Feelings Questionnaire (Parent-Report) | MFQ_P | 34 | nan | 2 | 0.1 |
| Mood and Feelings Questionnaire (Self-Report) | MFQ_SR | 33 | nan | 2 | 0.1 |
| The Positive and Negative Affect Schedule | PANAS | 20 | 2 | 2 | 0.05 |
| Repetitive Behaviors Scale | RBS | 43 | 5 | 3 | 0.1 |
| Screen for Child Anxiety Related Disorders (Parent-Report) | SCARED_P | 41 | 5 | 3 | 0.1 |
| Screen for Child Anxiety Related Disorders (Self-Report) | SCARED_SR | 41 | 5 | 3 | 0.3 |
| Social Communication Questionnaire | SCQ | 40 | nan | 4 | 0.02 |
| Strength and Difficulties Questionnaire | SDQ | 33 | 9 | 6 | 0.05 |
| Social Responsiveness Scale (School Age) | SRS | 65 | 7 | 3 | 0.5 |
| The Strengths and Weaknesses Assessment of Normal Behavior Rating Scale for ADHD | SWAN | 18 | 2 | 3 | 0.02 |
| Symptom Checklist (Parent-Report) | SympChck | 63 | nan | 3 | 0.1 |
| Teacher Report Form (School Age) | TRF | 116 | 19 | 8 | 0.5 |
| Youth Self Report | YSR | 119 | 11 | 3 | 0.2 |

# H    FULL LIST OF TOP 10 QUESTIONS FROM FACTORIZING *CBCL* QUESTIONNAIRE

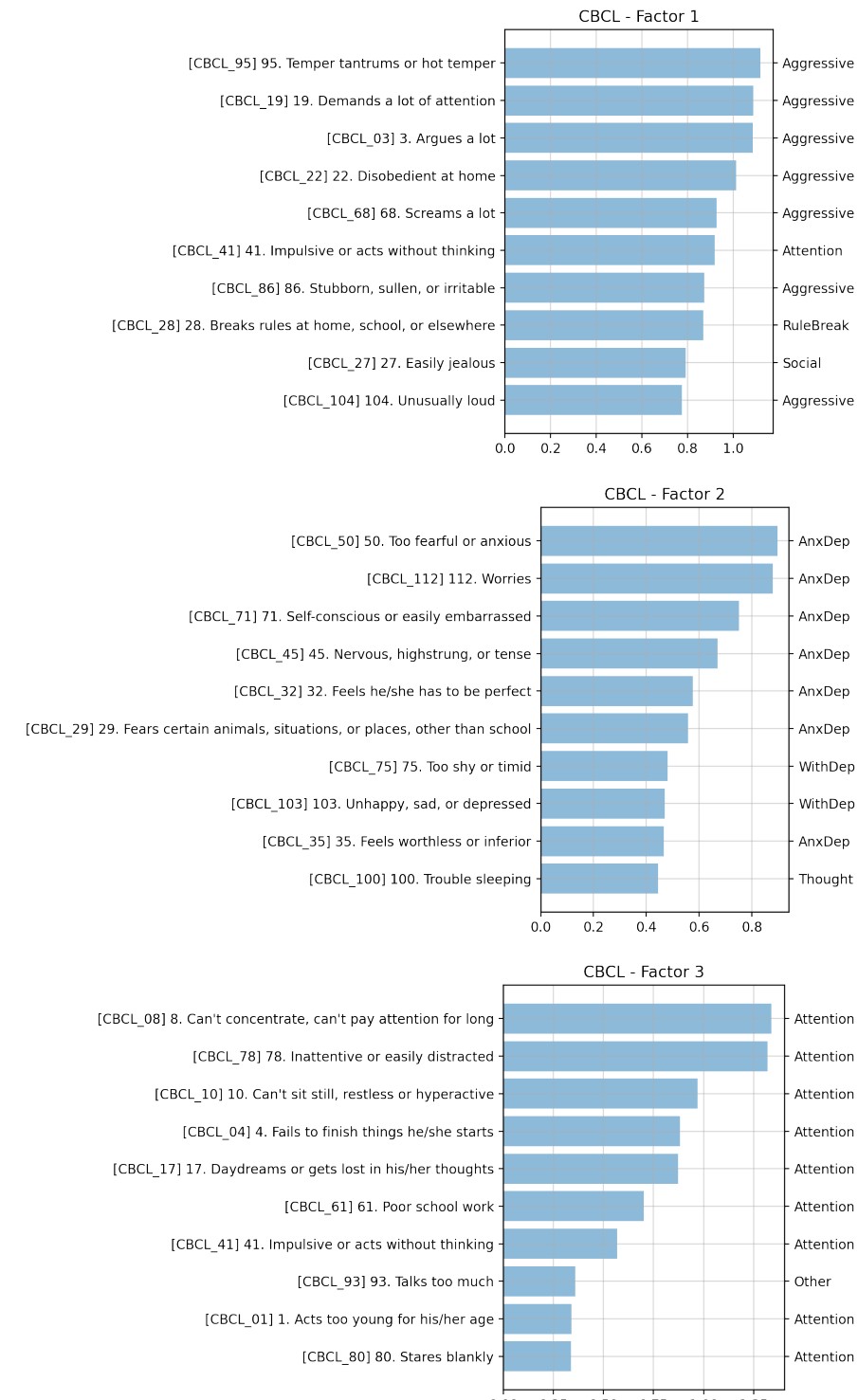

Figure 7: Top 10 questions ranked by $Q$ in *CBCL* using $Q$ obtained from ICQF (Factor 1-3).

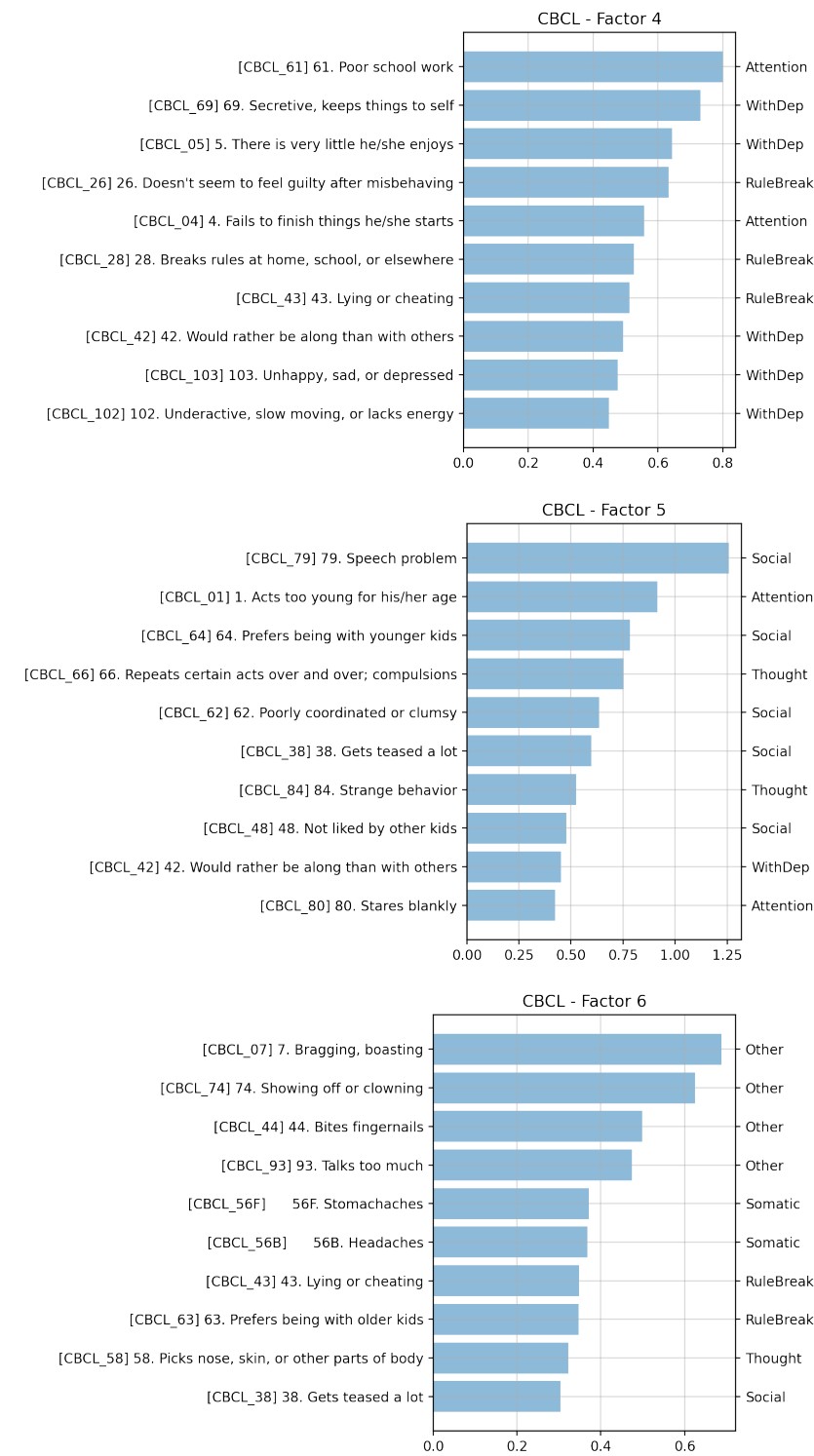

Figure 8: Top 10 questions ranked by **Q** in *CBCL* using **Q** obtained from ICQF (Factor 4-6).

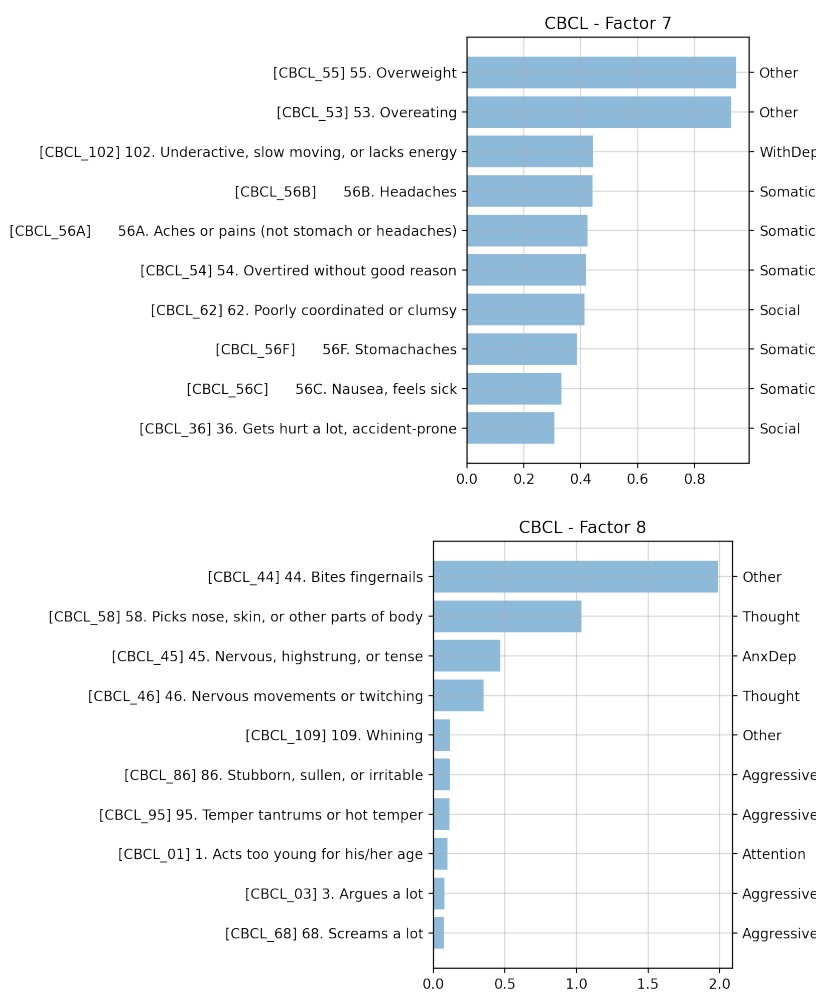

Figure 9: Top 10 questions ranked by $Q$ in *CBCL* using $Q$ obtained from ICQF (Factor 7-8).

# I  FULL LIST OF TOP 10 QUESTIONS FROM META-FACTORIZATION

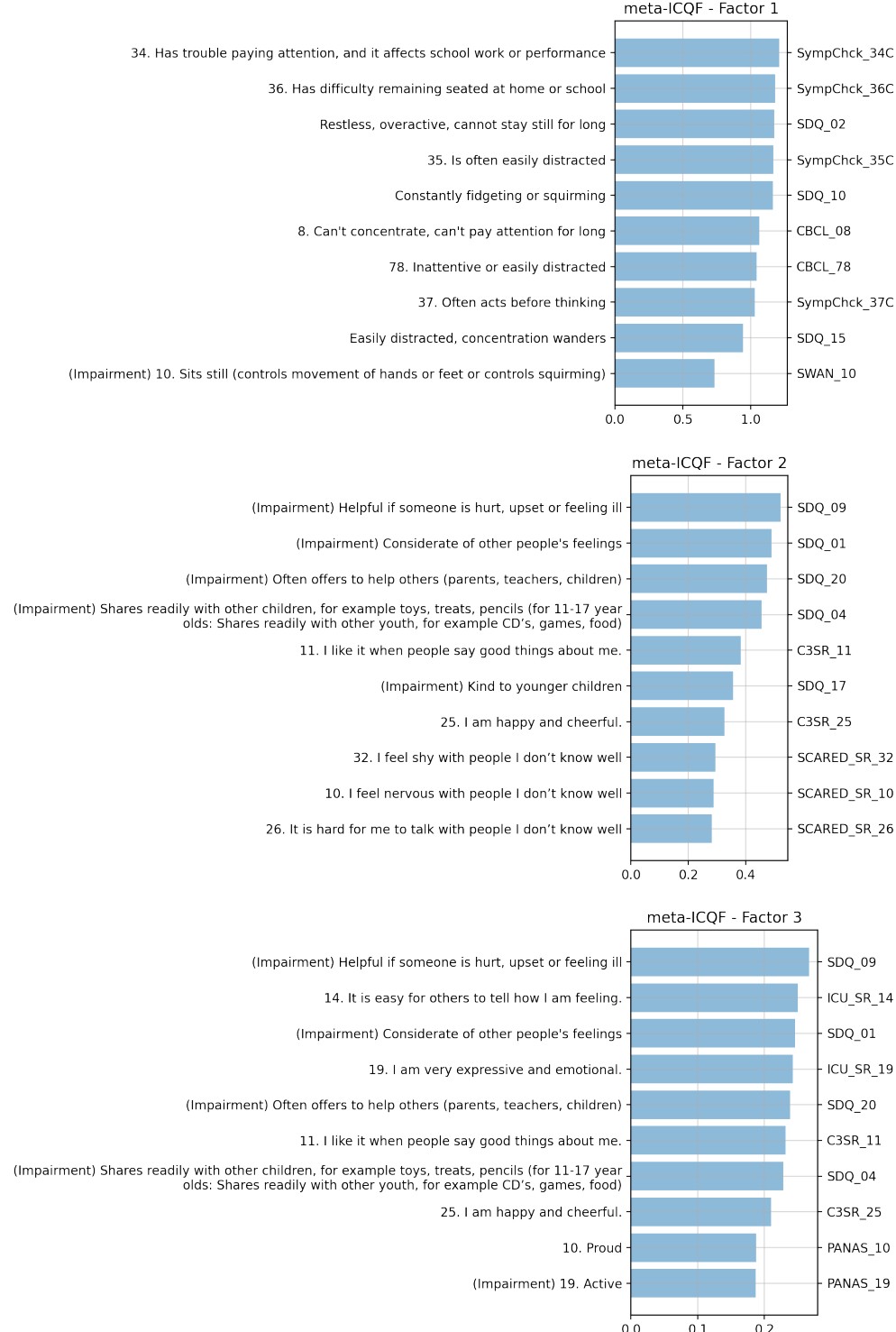

Figure 10: Top 10 questions ranked by **Q** in meta-ICQF (Factor 1-3).

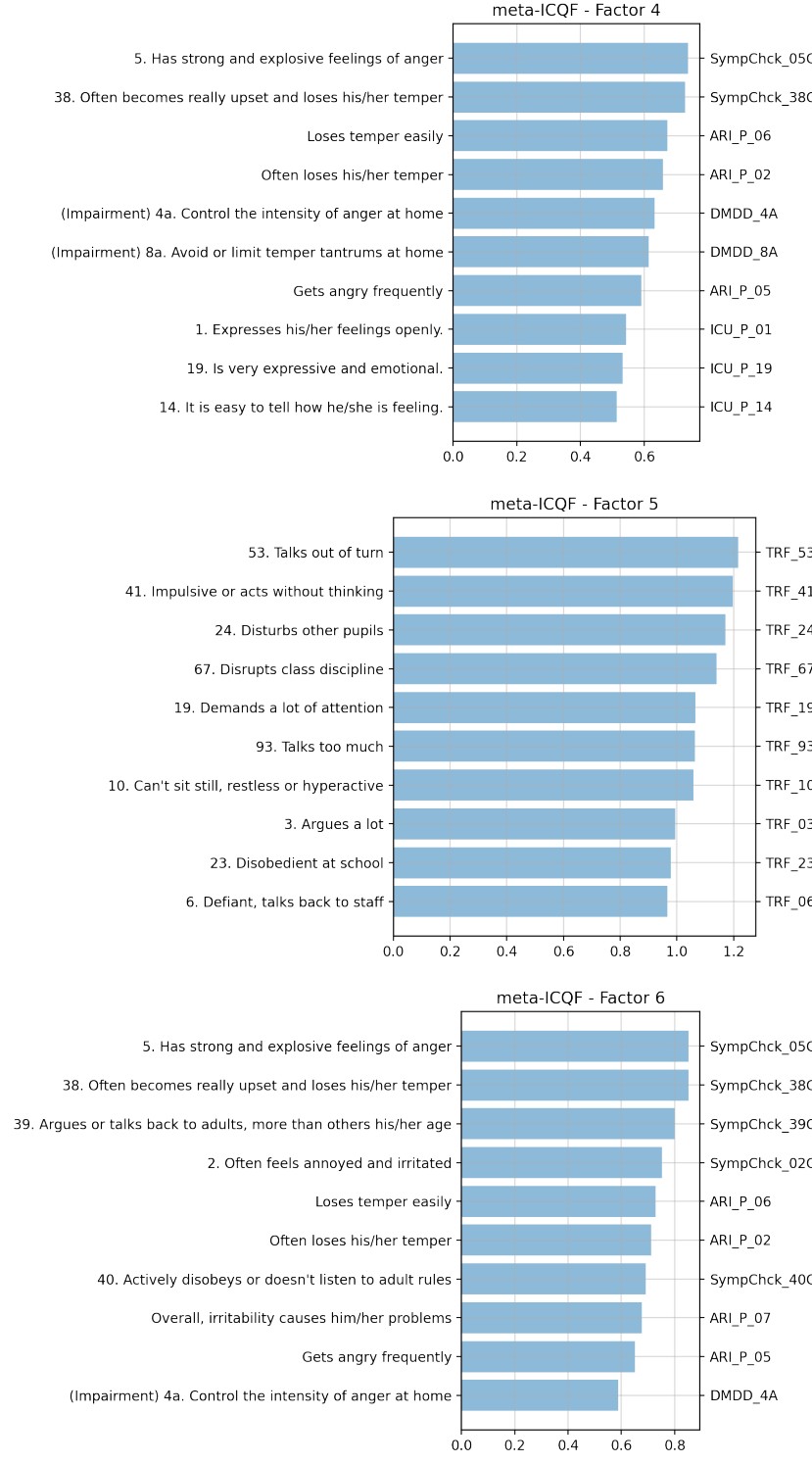

Figure 11: Top 10 questions ranked by $Q$ in meta-ICQF (Factor 4-6).

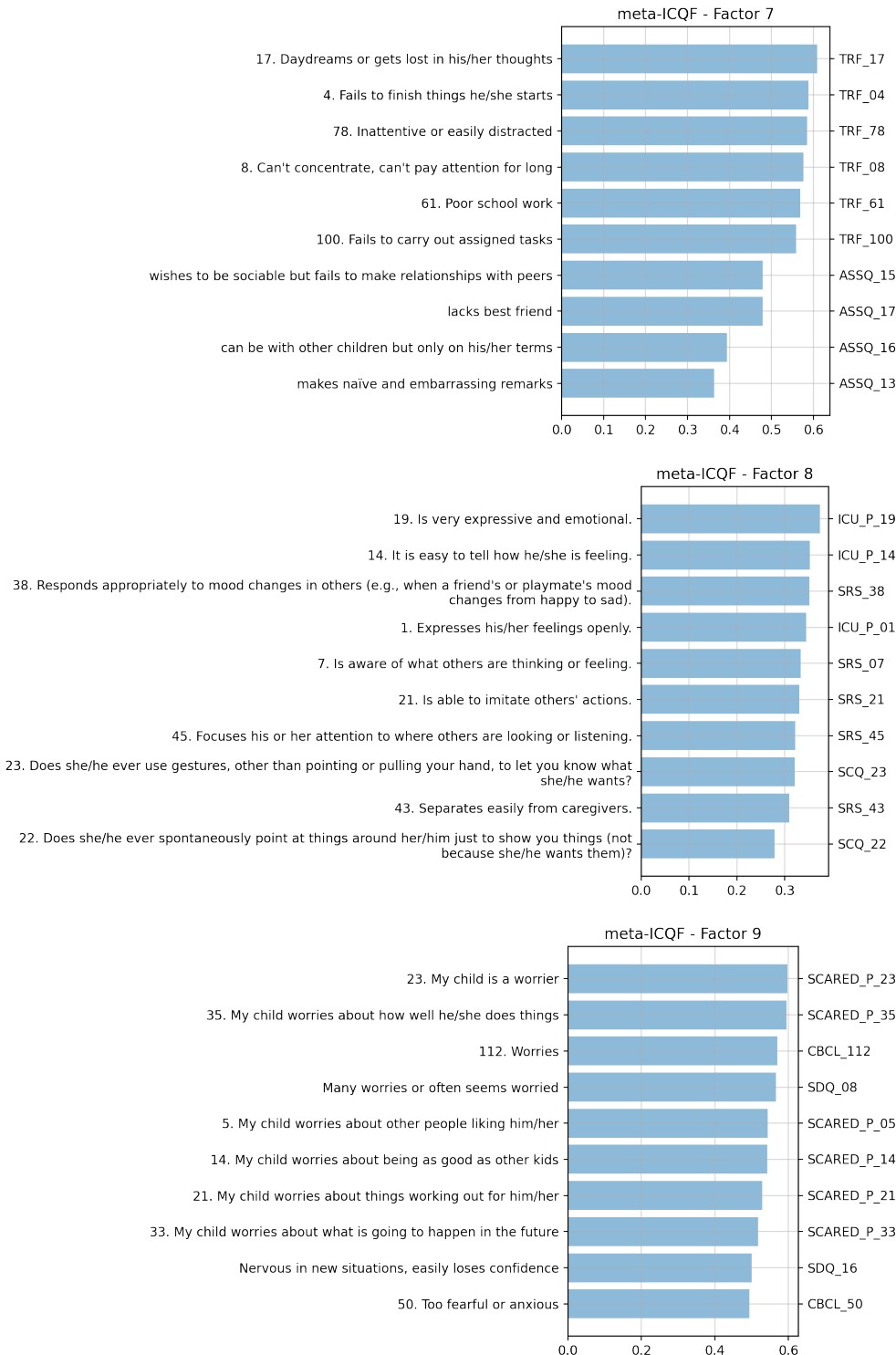

Figure 12: Top 10 questions ranked by $Q$ in meta-ICQF (Factor 7-9).

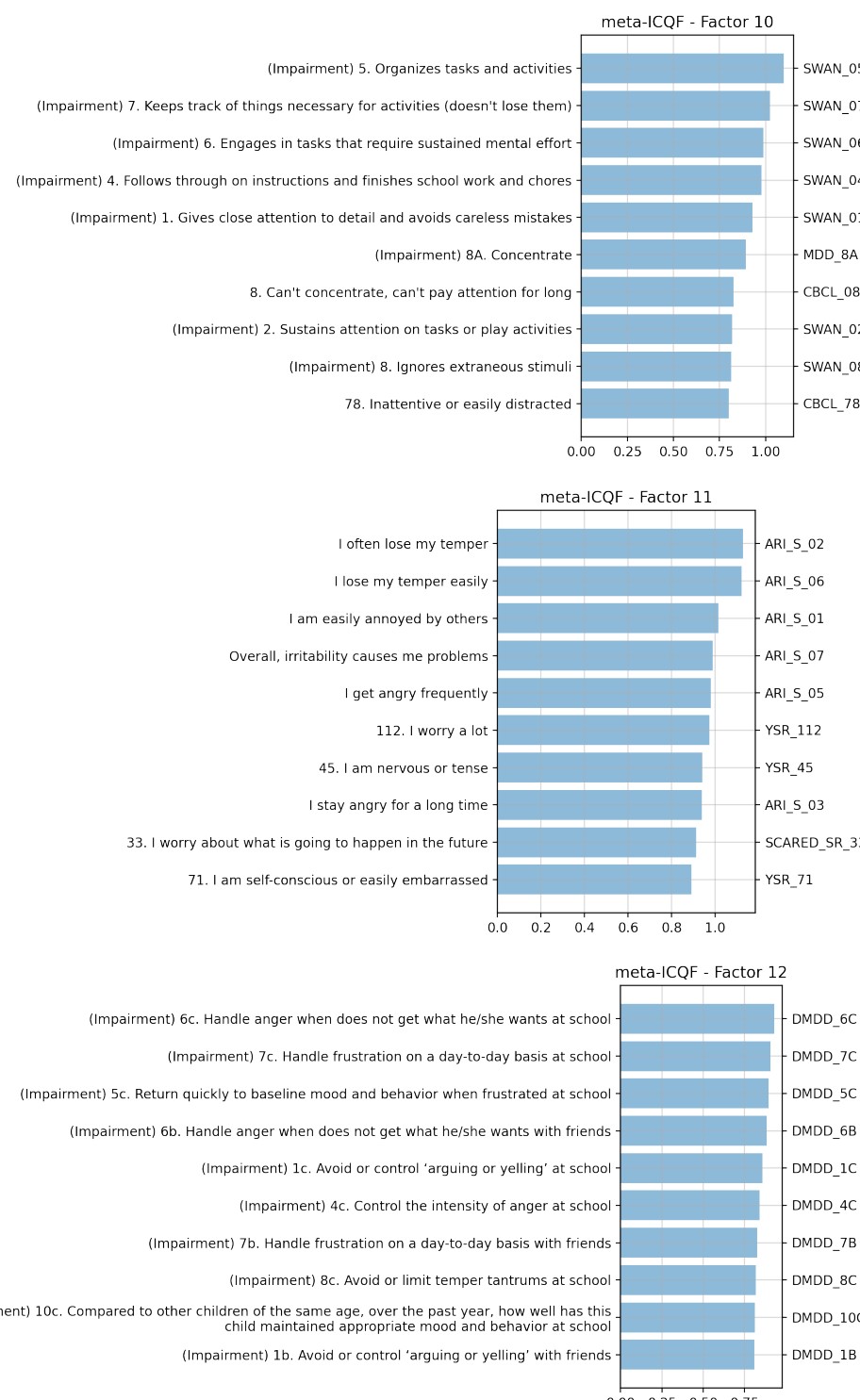

Figure 13: Top 10 questions ranked by $\boldsymbol{Q}$ in meta-ICQF (Factor 10-12).

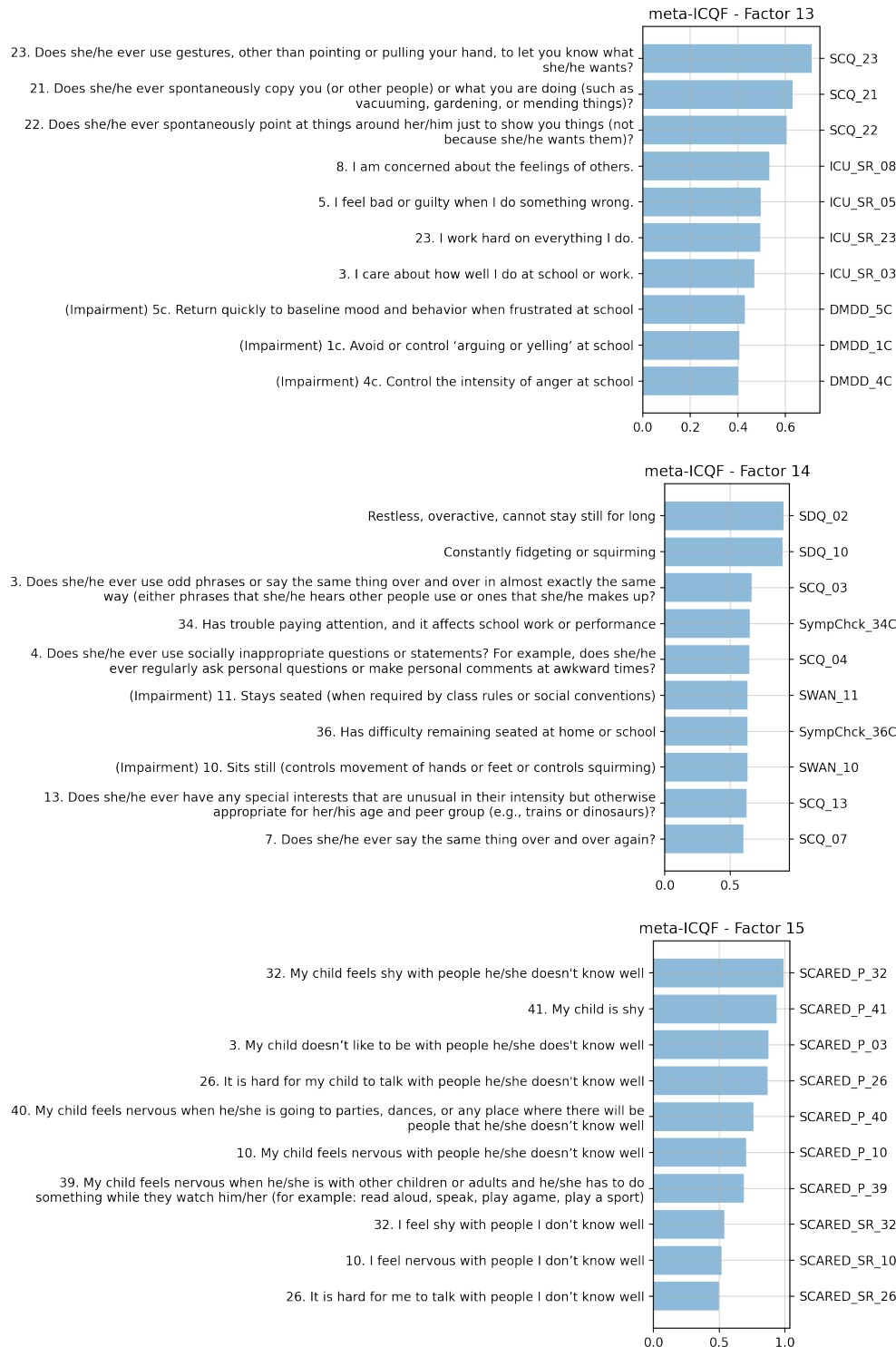

Figure 14: Top 10 questions ranked by $Q$ in meta-ICQF (Factor 13-15).

## J  ROC-AUC SCORES FOR EACH DIAGNOSTIC PREDICTION ACHIEVED BY DIFFERENT FACTORIZATIONS (EXPANSION OF TABLE 1

Table 5: ROC-AUC scores of the diagnostic prediction under different data inputs and factorizations.

| Diagnosis | Questionnaire | meta-ICQF | ICQF | $\ell_1$-NMF | FA-promax | subscales | raw |
|---|---|---|---|---|---|---|---|
| Depression | CBCL | – | 0.812 | 0.802 | 0.809 | 0.782 | 0.803 |
| | SDQ | – | 0.814 | 0.810 | 0.808 | 0.809 | 0.799 |
| | SympChck | – | 0.806 | 0.805 | 0.803 | NaN | 0.784 |
| | HBN | 0.829 | 0.823 | 0.806 | 0.813 | 0.813 | 0.818 |
| GenAnxiety | ″ | – | 0.823 | 0.825 | 0.825 | 0.798 | 0.816 |
| | | – | 0.825 | 0.817 | 0.830 | 0.815 | 0.817 |
| | | – | 0.759 | 0.798 | 0.794 | NaN | 0.778 |
| | | 0.840 | 0.833 | 0.832 | 0.836 | 0.832 | 0.840 |
| ADHD | ″ | – | 0.803 | 0.806 | 0.809 | 0.791 | 0.806 |
| | | – | 0.806 | 0.804 | 0.805 | 0.799 | 0.798 |
| | | – | 0.809 | 0.812 | 0.811 | NaN | 0.805 |
| | | 0.831 | 0.828 | 0.827 | 0.833 | 0.798 | 0.826 |
| Suspected ASD | ″ | – | 0.834 | 0.837 | 0.842 | 0.783 | 0.849 |
| | | – | 0.800 | 0.801 | 0.814 | 0.807 | 0.808 |
| | | – | 0.793 | 0.836 | 0.831 | NaN | 0.849 |
| | | 0.891 | 0.898 | 0.891 | 0.903 | 0.873 | 0.903 |
| Panic, Agoraphobia, Separation Anxiety, Social Anxiety | ″ | – | 0.766 | 0.755 | 0.751 | 0.742 | 0.767 |
| | | – | 0.742 | 0.737 | 0.738 | 0.746 | 0.739 |
| | | – | 0.750 | 0.754 | 0.760 | NaN | 0.753 |
| | | 0.802 | 0.786 | 0.785 | 0.790 | 0.770 | 0.789 |
| BPD | ″ | – | 0.776 | 0.769 | 0.785 | 0.780 | 0.780 |
| | | – | 0.767 | 0.769 | 0.767 | 0.759 | 0.777 |
| | | – | 0.750 | 0.749 | 0.756 | NaN | 0.754 |
| | | 0.786 | 0.777 | 0.775 | 0.775 | 0.777 | 0.784 |
| Specific Phobia | ″ | – | 0.623 | 0.614 | 0.632 | 0.621 | 0.647 |
| | | – | 0.663 | 0.654 | 0.663 | 0.657 | 0.737 |
| | | – | 0.658 | 0.662 | 0.648 | NaN | 0.656 |
| | | 0.654 | 0.669 | 0.650 | 0.628 | 0.643 | 0.661 |
| OCD | ″ | – | 0.790 | 0.767 | 0.775 | 0.778 | 0.755 |
| | | – | 0.740 | 0.744 | 0.742 | 0.728 | 0.657 |
| | | – | 0.765 | 0.786 | 0.801 | NaN | 0.818 |
| | | 0.782 | 0.722 | 0.764 | 0.738 | 0.722 | 0.783 |
| Eating Disorder | ″ | – | 0.635 | 0.626 | 0.599 | 0.586 | 0.602 |
| | | – | 0.651 | 0.631 | 0.620 | 0.610 | 0.737 |
| | | – | 0.633 | 0.637 | 0.606 | NaN | 0.610 |
| | | 0.665 | 0.637 | 0.665 | 0.697 | 0.645 | 0.694 |
| PTSD Trauma | ″ | – | 0.799 | 0.767 | 0.749 | 0.710 | 0.767 |
| | | – | 0.770 | 0.735 | 0.741 | 0.745 | 0.633 |
| | | – | 0.768 | 0.746 | 0.727 | NaN | 0.645 |
| | | 0.778 | 0.784 | 0.715 | 0.717 | 0.732 | 0.734 |
| Sleep problems | ″ | – | 0.787 | 0.787 | 0.825 | 0.770 | 0.803 |
| | | – | 0.805 | 0.800 | 0.802 | 0.796 | 0.782 |
| | | – | 0.797 | 0.789 | 0.787 | NaN | 0.761 |
| | | 0.850 | 0.841 | 0.832 | 0.828 | 0.826 | 0.839 |

## K    EVALUATION AND MODEL INFERENCE IN *CBCL–ABCD* DATASET

### K.1    DATA

Data from the Adolescent Brain Cognitive Development[SM] (*ABCD*) study (https://abcdstudy.org), held in the NIMH Data Archive (NDA), consists of over 11,000 youth age 9–10. They are recruited from 21 sites across the US, aiming to provide a socio-demographically-diverse sample. Psychopathology was examined in the children using the parent-reported Child Behavior Checklist (*CBCL*). Similarly, we have the age and sex at birth of each participants as confounds and diagnostic labels (Table 6) for 11 conditions used for the model evaluation. By eliminating subjects with incomplete age and sex information, we obtain 11,681 unique subjects in *CBCL–ABCD* questionnaire.

Table 6: Number of positive diagnostic labels in *ABCD* dataset

| Diagnosis | number of positive cases |
|---|---|
| Depression | 743 |
| GenAnxiety | 715 |
| ADHD | 2550 |
| Suspected ASD | 3244 |
| Panic, Agoraphobia, Separation Anxiety, Social Anxiety | 2031 |
| BPD | 802 |
| Specific phobia | 3131 |
| OCD | 1258 |
| Eating Disorder | 1205 |
| PTSD Trauma | 550 |
| Sleep problems | 1129 |

### K.2    DIAGNOSTIC PREDICTION (PARALLEL TO SECTION 5.3.2)

We perform the same diagnostic prediction evaluation on *CBCL–ABCD* dataset as reported in Section 5.3.2. We split participants into train, validation and test sets with ratio $70/15/15$, based on the distribution of confounds and diagnostic labels. We resample 50 dataset splits using different seeds and carry out prediction experiment in each split. Table 7 reports the summary of averaged AUCs achieved by different factorization methods.

Table 7: Averaged ROC-AUC scores of the diagnostic prediction in *CBCL–ABCD* dataset

| Diagnosis | Factorization | | | | |
|---|---|---|---|---|---|
| | ICQF | $\ell_1$-NMF | FA-promax | subscales | raw |
| ADHD | 0.914 | 0.913 | 0.912 | 0.907 | 0.915 |
| BPD | 0.781 | 0.776 | 0.781 | 0.775 | 0.783 |
| Depression | 0.787 | 0.785 | 0.786 | 0.794 | 0.791 |
| Eating Disorder | 0.546 | 0.551 | 0.543 | 0.555 | 0.567 |
| GenAnxiety | 0.867 | 0.830 | 0.872 | 0.868 | 0.884 |
| OCD | 0.789 | 0.748 | 0.788 | 0.781 | 0.805 |
| PTSD Trauma | 0.804 | 0.803 | 0.812 | 0.798 | 0.814 |
| Panic, Agoraphobia, Separation Anxiety, Social Anxiety | 0.767 | 0.727 | 0.769 | 0.765 | 0.777 |
| Sleep problems | 0.825 | 0.726 | 0.762 | 0.743 | 0.833 |
| Specific Phobia | 0.684 | 0.654 | 0.680 | 0.681 | 0.695 |
| Suspected ASD | 0.777 | 0.755 | 0.774 | 0.769 | 0.782 |
| Average | 0.776 | 0.752 | 0.771 | 0.767 | 0.786 |

The null hypothesis is rejected and the post-hoc Nemenyi test indicates that *subscales* and $\ell_1$-NMF are slightly but significantly worse than the *raw* setting, while other differences are inconclusive. Similarly, we conclude that ICQF preserves diagnostic information even though we introduce additional regularization and constraints versus other methods.

### K.3 STABILITY OF MODEL

In Section 5.3.3, we demonstrated that the additional regularization and constraints in ICQF improve the interpretability of the factorization. Coherently, we should expect to obtain a more consistent factorization result from ICQF if it is applied on dataset from another population. Using *CBCL–ABCD* dataset, we extend our evaluation via studying the stability of factor loadings (matrix $Q$) in a cross-population scenario.

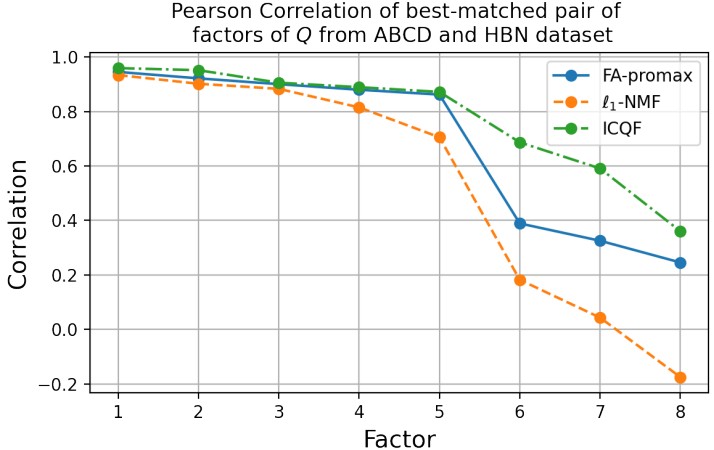

Figure 15: Sorted Pearson correlation of best-matched pair of factor loadings $Q$ obtained from *CBCL–ABCD* and *CBCL–HBN* questionnaire.

For each method (ICQF, $\ell_1$-NMF and FA-promax), we factorize *CBCL–HBN* (care-seeking population) and *CBCL–ABCD* dataset (general-diversified population) separately to obtain the factor loadings $Q_{HBN}$ and $Q_{ABCD}$ respectively. In this experiment, for fairness sake, we set the number of factors $k = 8$ for all methods. Considering $Q_{HBN}$ as the reference, we perform the greedy-matching algorithm to re-order factors in $Q_{ABCD}$ and compute the Pearson correlation coefficient for every best-matched pair of factors from $(Q_{HBN}, Q_{ABCD})$.

Figure 15 shows the trend of the 8 Pearson correlation coefficients achieved by ICQF, $\ell_1$-NMF and FA-promax factorizations. Since there is a sign ambiguity of FA-promax, we take an absolute to all correlation coefficients achieved from FA-promax. The first 5 correlation coefficients are close to 1 for all three methods, suggesting that all methods capture similar factors in two different populations. However, starting at the 6[th] pair of factors, correlation coefficient achieved by ICQF is superior to those from $\ell_1$-NMF and FA-promax. This indicates that the extra constraints help stabilize the factorization outcomes and are potentially advantageous to transfer learning or model interpretation.

## L    ROC-AUC TREND OF OTHER DIAGNOSES IN $I$-META-QUESTIONNAIRE

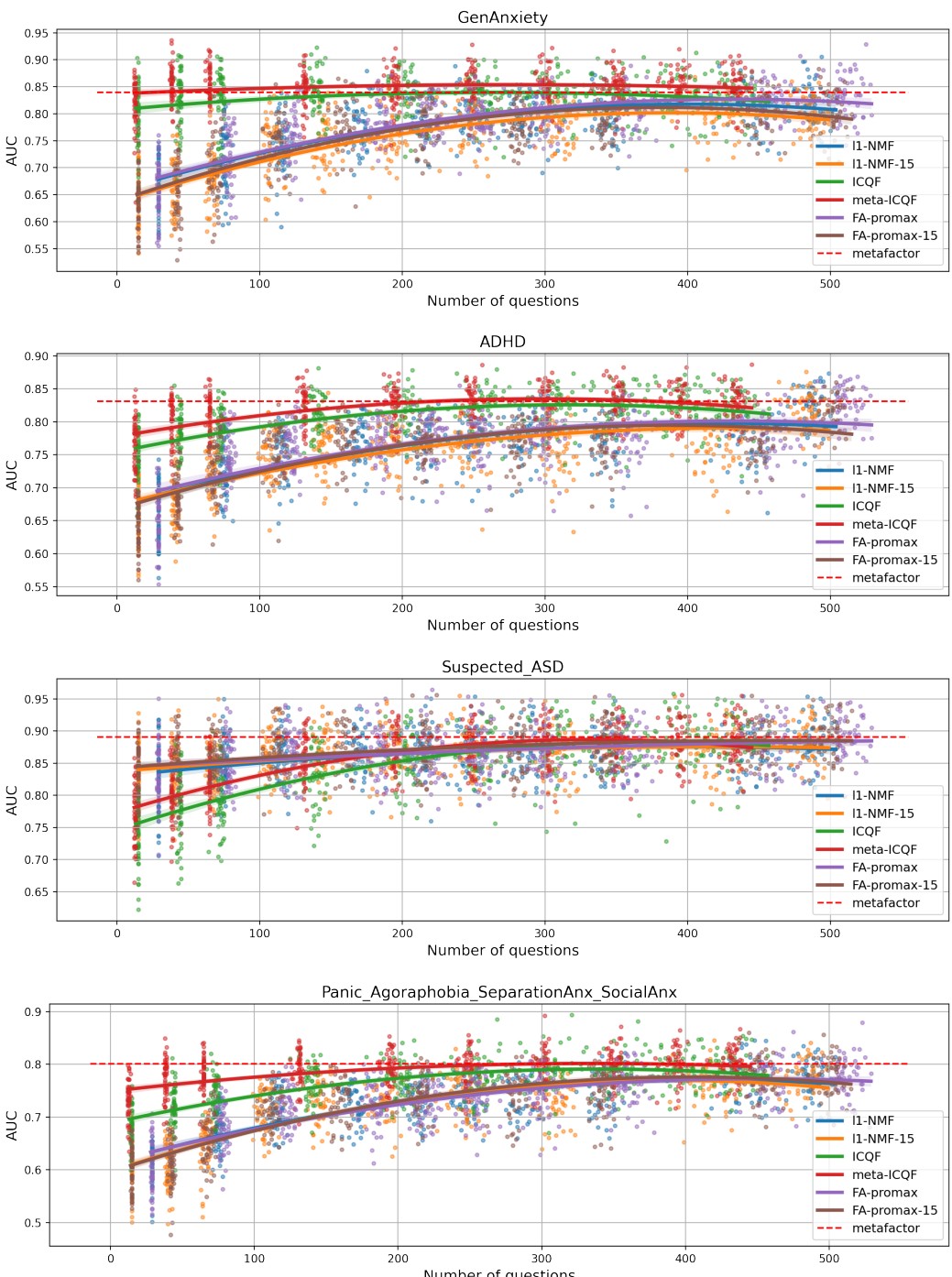

Figure 16: ROC-AUC trend of GenAnxiety, ADHD, Suspected ASD and {Panic, Agoraphobia, SeparationAnx, SocialAnx}.

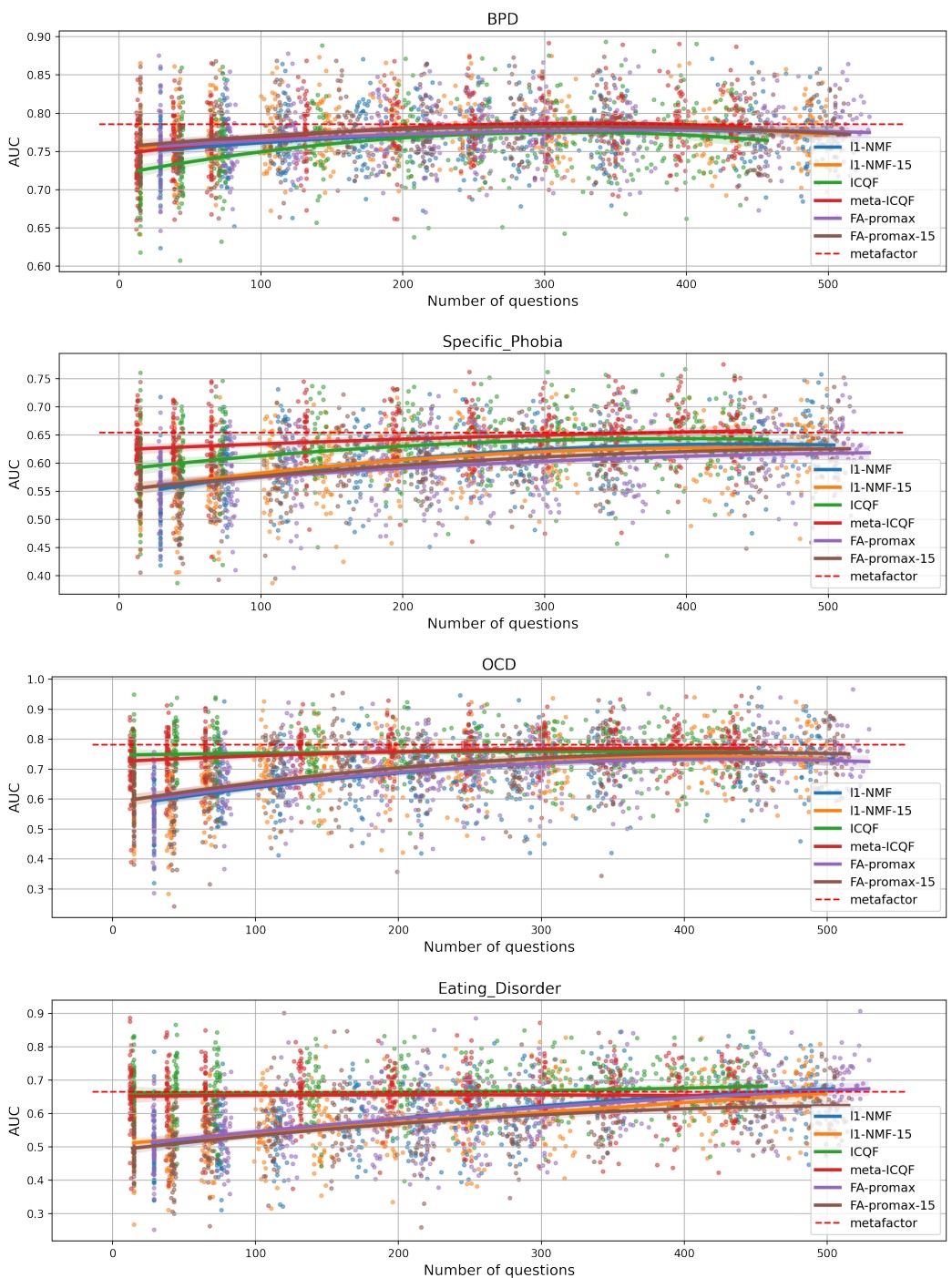

Figure 17: ROC-AUC trend of BPD, Specific Phobia, OCD and Eating Disorder.

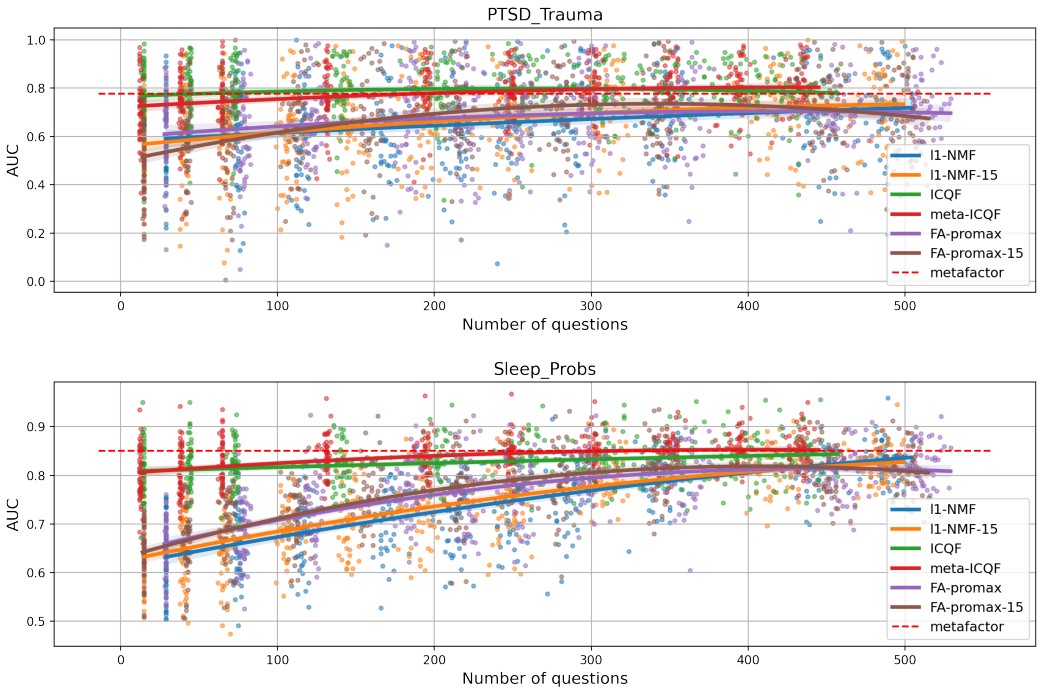

Figure 18: ROC-AUC trend of PTSD Trauma and Sleep Problems

