# OpenReview forum: "Interpretable (meta)factorization of clinical questionnaires to identify general dimensions of psychopathology"
_ICLR.cc/2023/Conference — Submitted to ICLR 2023_

### Official Review · Reviewer_R2ij · 2022-10-24

**Confidence:** 4
**Correctness:** 3
**Technical Novelty And Significance:** 2
**Empirical Novelty And Significance:** 2
**Recommendation:** 3

**Clarity, Quality, Novelty And Reproducibility:**

Clarity is satisfactory and the proposed algorithm is clearly presented.

As explained in "Strengths and Weaknesses" section, the novelty of the proposed algorithm for matrix factorization (a well developed field) is considered incremental.

**Strength And Weaknesses:**

Strengths:
+ The application scenario is clearly presented.
+ The problem is clearly formulated in general.

Weaknesses:
- The key contribution of the paper is not clear. As mentioned in the paper, there are different alternatives to organize the multiple questionnaire data matrices for the factorization. It seems that performance gained in terms of ROC-AUC together with the interpretability is due to the application of constraints to factor values and reconstructed matrices. Is it not clear what happen if similar tricks are applied to the setting with the questionnaire data matrices concatenated. Also, the way to handle missing values and the use of ADMM for constrained optimisation is not new. Overall speaking, the novelty in term of algorithm is considered incremental.
- It seems that the interpretability could be related to the control of the sparsity as discussed in Section 3.1. I noticed only that the values of p and q for controlling the sparsity are set to be 1. There seems no more investigation on sparsity afterwards.

Question:
In Section 3.1, regarding continuous confound, if one wants to consider different ranges of value, will the mirroring procedure be sufficient?

Some typos
* The variables p and q are used twice to denote different concepts, sparsity control and row and column shuffling in BCV (Section 3.1).
* The mask defined in ICQF is not the same as the mask defined in 3.3, but the same notation is used.

**Summary Of The Paper:**

The paper proposes an algorithm to identify "meta-factor" from subjects' answers to multiple questionnaires with different lengths. Specific constraints are introduced and the problem is formulated a constrained optimization problem. An ADMM algorithm is derived to compute the factorization solution. Performance comparison with some existing methods have been presented and the way of using the proposed method to interpret the meta-factors extracted from the questionnaires' results was illustrated.

**Summary Of The Review:**

This paper proposed a matrix factorization algorithm which can handle multiple questionnaire data matrices together with the objective to extract some meta-factor to support psychiatry research. While the paper is clearly motivated and presented from the perspective of application, the novelty from the perspective of machine learning algorithms is considered incremental.

---

> ### Author Response · Authors · 2022-11-18
> **Detail response of individual comments and questions**
>
> We would like to thank the reviewer for taking the time to review the paper and raising important questions. In the following, we will address individual comments and questions raised by the reviewer. Due to the restriction of number of characters, our response is split into different comment posts.
>
> - "The key contribution of the paper is not clear"
>     > We apologize if this was not clear. We ask the reviewer to please refer to the shared response to all reviewers, given that this point was raised multiple times.
>
> - "It seems that performance gained in terms of ROC-AUC together with the interpretability is due to the application of constraints to factor values and reconstructed matrices."
>     > The reviewer is correct. We use performance in the diagnostic prediction task primarily as an indicator that our method preserves diagnostic information, even as it imposes constraints that also increase interpretability. We discuss this in more detail in the shared response to all reviewers.
>
> - "Is it not clear what happen if similar tricks are applied to the setting with the questionnaire data matrices concatenated."
>     > As suggested by the reviewer, we included these as new results in the revised manuscript. Specifically, Table 1 now has an ICQF result for the HBN concatenated questionnaires. The average performance (0.782) is close to that obtained with the meta-ICQF (0.792). These results hold for individual diagnostics, please see Appendix J. We also included a line for ICQF in Figure 3, again applying the method to concatenated questionnaires rather than doing a meta-factorization. The results are slightly worse than for meta-ICQF in this diagnosis, but still better than those of the other baselines. We did the same for other diagnoses, and meta-ICQF is substantially better in some of them; for more details, please see the figures in Appendix L.
>     >
>     > A different dimension of assessment would be a subjective comparison of the interpretability of the meta-factors obtained with meta-ICQF, and the factors obtained with ICQF on concatenated questionnaires. The former yields a model with 15 meta-factors, and the latter one with 22 factors. After inspection, some of the 22 factors proved to be redundant. In contrast, the top 10 questions for each of the 15 meta-factors were very coherent, and very distinct across them.
>
> - "It seems that the interpretability could be related to the control of the sparsity as discussed in Section 3.1. I noticed only that the values of $p$ and $q$ for controlling the sparsity are set to be 1. There seems no more investigation on sparsity afterwards."
>     > The reviewer is correct. Sparsity is one factor in interpretability, but note that the other constraints also play a role. For example, constraining factor loadings over questions to be in the same range as question answers allows them to be interpreted as a factor-specific answer pattern. Restricting factors to be between 0 and 1 makes them interpretable as a degree of presence.
>     >
>     > In more detail, the optimal choice for sparsity would be the $L_0$ nuclear norm ($p=q=0$). Given the flexibility of our formulation, it is possible to consider nuclear norm regularization, using an algorithm such as that in (Oh et al. 2015) for sub-problems 1 and 2. However, this is computationally expensive, as it requires using singular-value thresholding. Since sample size or question numbers may be large, there may be scalability issues, and hence we only focus on sparsity regularization in this work.
>     >
>     > Given this, and the fact that $L_1$ norm is a good relaxation of the $L_0$ nuclear norm (it is convex),
>     we use the standard sparsity control ($p = q = 1$). For our purposes, this was sufficient; however, our formulation makes it possible to consider other norms, which might be relevant in other applications.
>     For general $L_p$ regularization, the ADMM approach allows easy modification of the algorithm. For instance, setting $p=2$ changes sub-problem 1 and 2 into constrained ridge regression problems, which can also be solved efficiently by, e.g. quadratic programming. For higher order regularization, the regularization may not have an intuitive interpretation.
>
> - "if one wants to consider different ranges of value, will the mirroring procedure be sufficient?"
>     > This is a good question. At the point that we fit the model, the confound variables are all given. We used the range of values present in the dataset we were working with, but there is no obstacle to using a different, wider range (e.g. derived from the population). If confounds are continuous, the mirroring step is the same as for categorical ones. Alternatively, we can also split the confounds into quantile buckets, if it might make more sense in specific applications.

---

> > ### Author Response · Authors · 2022-11-18
> > **Detail response of individual comments and questions**
> >
> > - "The variables $p$ and $q$ are used twice to denote different concepts, sparsity control and row and column shuffling in BCV (Section 3.1)."
> >     > Thank you very much for pointing out the misleading notation. We changed the $p, q$ notation in BCV to $b_r$ and $b_c$ in the revised manuscript.
> >
> > - "The mask defined in ICQF is not the same as the mask defined in 3.3, but the same notation is used."
> >     > We apologize if this was unclear. In ICQF (single questionnaire setting), we use $\mathcal{M}$  to denote the mask matrix; a zero entry means a missing response for the corresponding participant and question. In meta-ICQF (multi-questionnaire setting), we would refer to this mask for questionnaire $i$ as $\mathcal{M}_i$. We  then use $\mathcal{E}_i := \mathcal{D}_i \cdot \mathcal{M}_i$ to represent missing entries for a participant in the factorization of questionnaire $i$.
> >
> >
> > Reference:
> > - Oh, T. H., Matsushita, Y., Tai, Y. W., & So Kweon, I. “Fast randomized singular value thresholding for nuclear norm minimization”. Proceedings of the IEEE Conference on Computer Vision and Pattern Recognition, pp. 4484–4493, 2015.

---

> > > ### Comment · Reviewer_R2ij · 2022-12-03
> > > **Response**
> > >
> > > The authors' effort to address my comments is highly appreciated. Yet the originality and key contribution is still not clear to me.

---

> > > > ### Author Response · Authors · 2022-12-04
> > > > **The originality and key contribution**
> > > >
> > > > We thank the reviewer for taking the time to read through our
> > > > response, as well as the shared response to all reviewers. If we may, we will provide a succinct reply to your question that adds to the points already raised.
> > > >
> > > > The **originality** of this work stems primarily from identifying a specific need of psychiatry and psychology researchers -- interpretable factorization of questionnaires -- and translating it into formal requirements that a factorization method must satisfy. Beyond that, the specific application of the method to meta-factorization -- identifying the main factors of psychopathology -- is intrinsically important for psychiatry research. It also leads to the idea of being able to synthesize short, informative questionnaires from the meta factorization, which is practically useful for clinical applications.
> > > >
> > > > From this perspective, the **key contributions** of this paper are identifying these needs or potential applications, developing a new method that satisfies all the requirements for addressing them -- as we could find none that did -- and carrying out a thorough evaluation of whether they were addressed.
> > > >
> > > > Technically, we aimed for the simplest, yet flexible, method that would satisfy all the requirements, and for which we could also provide a proof of convergence and formal guarantees for the quality of the solution. We also aimed to make the method friendly to domain users, by supporting automated estimation of the number of factors, minimizing the number of hyper-parameters, and supporting imputation. From a narrow technical perspective, we agree this is an incremental development in matrix factorization. Even then, we think the same could be said for many factorization methods with novel constraints,  when they were first published.
> > > >
> > > > The question the reviewer may then ask is whether ICLR is a good venue for work with this contribution profile. We believe that it is, and this is why we chose the submission area "**Machine Learning for Sciences (e.g. biology, physics, health sciences, social sciences, climate/sustainability)**". This specific method enables scientific discovery in applications where questionnaires are the primary data type, which are ubiquitous across psychology
> > > > and the social sciences in general. From that perspective, we think this is clearly a "ML for Science" project, and we would ask that it be considered both with respect to the scientific applications we present, and the many others it could potentially be deployed in. We again thank the reviewer for giving us the opportunity to elaborate on this point.

---

### Official Review · Reviewer_V6jc · 2022-10-26

**Confidence:** 5
**Correctness:** 4
**Technical Novelty And Significance:** 2
**Empirical Novelty And Significance:** 2
**Recommendation:** 3

**Clarity, Quality, Novelty And Reproducibility:**

The manuscript is reasonably clear and I see no major error.

However, I do not see what make it stand out from a vast literature of prior work developing matrix factorization or dictionary learning with various applications, including brain imaging.

In terms of reproducibility, I do not believe that code has been shared.

**Strength And Weaknesses:**

The latent factors seem indeed to reveal some interesting structure in the questionnaires.

However, the method is very classic and many variants of such matrix factorization have been published.

Also, in terms of solver, the alternated minimization are very slow. Online methods are much faster, such as the original work of Mairal and later work adapting it to me complex settings (including brain imaging).

**Summary Of The Paper:**

On the context of interpreting questionnaires in psychology, this submission contributes a method for non-negative matrix factorization with additional constraints to further enhance the interpretability of factors.

The approach uses penalties on both matrices of the factorization and solves the minimization problem by alternated minimization with inner problems solved by ADMM.

The method is demonstrated on a psychology cohort dataset, looking at diagnostic performance with prediction from latent factors.

**Summary Of The Review:**

An interesting idea but that does not stand out as very different from prior efforts.

---

> ### Author Response · Authors · 2022-11-18
> **Detail response of individual comments and questions**
>
> We would like to thank the reviewer for taking the time to review the paper. In the following, we will address individual comments and questions raised by the reviewer. Due to the restriction of number of characters, our response is split into different comment posts.
>
> - "However, the method is very classic and many variants of such matrix factorization have been published."
>     > We agree with the reviewer that there are many variants of matrix factorization, and even of non-negative matrix factorizations (e.g. see the Advanced Matrix Factorization Jungle website https://sites.google.com/site/igorcarron2/matrixfactorizations).
>     We developed our method because we could find no other non-negative factorizatization that would control bounds of both factor and reconstructed matrix, together with sparsity constraints. The constraints were defined in discussion with our clinical collaborators, to reflect characteristics they felt would increase interpretability of a factorization. The bounded constraints on factors $W$ also allow a direct extension of ICQF to meta-ICQF for in the multi-questionnaire setting. We provided additional details on our motivation in the shared response, as other reviewers had similar questions.
>
> - "the alternated minimization are very slow. Online methods are much faster, such as the original work of Mairal and later work adapting it to me complex settings (including brain imaging)."
>     > We agree that using online methods should, in principle, reduce the computational burden. This said, and given that the whole factorization takes about one minute for the typical questionnaire (thousands of participants, tens to hundreds of questions), we did not feel it was necessary to optimize further at the cost of complicating our algorithm (and convergence proofs). In addition, as we are focusing primarily on the quality and interpretability of the factorization, we felt that including the $\ell_1$-NMF optimized with block coordinate descent method would be a reasonable baseline to compare against for our study.
>     >
>     > We are familiar with Mairal's work, and have added (Mairal et al. 2010) to the related work section in our revised paper as a possible avenue for improving performance, if necessary. We note that Algorithm 1 in Mairal et al. work is also an alternated minimization approach, just like our method. The other differences are using Least Angle Regression (LARS) algorithm for minimizing the LASSO problem, and using a stochastic optimization approach.
>     >
>     > A key reason for using ADMM for sub-problems 1 and 2 is that we are dealing with non-negative and upper bound constraints (which are crucial for interpretability, as described in Section 5.3.3.). In other words, sub-problems 1 and 2 are constrained LASSO minimization problems. ADMM combined with FISTA is a good optimization choice in this context, as discussed in (Gaines, Kim, and Zhou 2018).
>     >
>     > Using online methods instead of ADMM would be feasible if we were just handling one questionnaire with a large number of participants. In a multi-questionnaire setting, however, participants may not respond to all questionnaires. As a consequence, the data distribution (denoted as $p(x)$ in Algorithm 1 of Mairal's work) is challenging to estimate / de-bias when we generate batches $x$. In contrast with the brain imaging setting, missing entries have to be taken into consideration carefully if using a stochastic optimization approach (such as online learning). There is recent work (e.g. Sportisse et al. 2020) focusing on de-biasing SGD to handle data with missing values but, again, it would make our algorithm substantially more complicated without there being a clear need for the increase in performance.

---

> > ### Author Response · Authors · 2022-11-18
> > **Detail response of individual comments and questions**
> >
> > - "later work adapting it to me complex settings (including brain imaging)." and "However, I do not see what make it stand out from a vast literature of prior work developing matrix factorization or dictionary learning with various applications, including brain imaging."
> >     > We appreciate the reviewer raising this concern. We are very familiar with brain imaging data, and have published multiple supervised matrix factorization methods in that domain.  Our method is designed for use with questionnaires: there are typically tens to hundreds of variables, and they are non-negative, and relatively sparse. Functional brain imaging activation values are real-valued (even if bound), and are certainly not sparse; the same is true for structural brain imaging measures. Even if sample sizes are comparable to our situation, in terms of number of participants, the number of voxels or vertices is orders of magnitude larger ($O(10K)$ to $O(100K)$). We did not design this method to deal with this particular type of data. We agree with the reviewer that performance matters, but it does need to be considered in a trade-off with other requirements such as the characteristics of the factorization or the dataset.
> >
> > Reference:
> > - Mairal, J., Bach, F., Ponce, J., & Sapiro, G. (2010). “Online learning for matrix factorization and sparse coding.” Journal of Machine Learning Research 11.1, 2010.
> > - Gaines, B. R., Kim, J., & Zhou, H. “Algorithms for fitting the constrained lasso”. Journal of Computational and Graphical Statistics 27.4, pp. 861–871, 2018.
> > - Sportisse, A., Boyer, C., Dieuleveut, A., & Josse, J. “Debiasing averaged stochastic gradient descent to handle missing values”. Advances in Neural Information Processing Systems 33, pp. 12957–12967, 2020.

---

> > > ### Comment · Reviewer_V6jc · 2022-11-24
> > > **The machine learning contribution feels minor**
> > >
> > > I understand that, probably, no specific algorithm was written with the specific set of constraints and penalties used here, but this is a generalization of very close methods using ADMM for matrix factorization, and the proofs carry over naturally.
> > >
> > > While I see that for the data at hand scalability is not an issue, it implies that the method is limited to small data and that the algorithmic contribution is minor.
> > >
> > > Given that ICLR is a fairly general machine learning conference, and not one focused on the type of data used in the submission, these limitations seem important ones.

---

> > > > ### Author Response · Authors · 2022-11-28
> > > > **Addressing some of the further points the reviewer raised**
> > > >
> > > > We thank the reviewer for taking the time to read our detailed response to their points and questions, as well as the shared response to all reviewers. We do believe we can address some of the further points the reviewer raised, so we will take the opportunity to do so, if we may.
> > > >
> > > > - "Given that ICLR is a fairly general machine learning conference, and not one focused on the
> > > > type of data used in the submission, these limitations seem important ones."
> > > > > We agree that ICLR is a general machine learning conference. This said, we believe that papers introducing methods for learning representations for specific applications are very much within scope. This is why we chose the submission area "**Machine Learning for Sciences (e.g. biology, physics, health sciences, social sciences, climate/sustainability)**".
> > > > >
> > > > > We would also like to note that questionnaires are a very common data type. Beyond clinical practice and research, already described in the paper, they are also used in psychology research. There, the goal may be to discover latent variables that explain aspects of behavior that can be coded numerically. Similar examples could be given for survey data in almost any social science.
> > > > >
> > > > > A different aspect of the contribution is making the method friendly to users in these areas. For example, we minimize the number of hyper-parameters, and provide a simple way to determine the remaining ones (e.g. the number of factors $k$ and sparsity penalty $\beta$). We transparently handle missing entries, and partial overlap of participants in multiple questionnaires.
> > > >
> > > > - "While I see that for the data at hand scalability is not an issue, it implies that the method is
> > > > limited to small data and that the algorithmic contribution is minor."
> > > > > We understand the concern of the reviewer, and should have been clearer on this point: a questionnaire dataset with thousands of participants and hundreds of questions is very large, across every application that we have encountered in psychiatry or psychology. Larger still would likely mean an increase in number of participants, rather than more questions. This is why we believe that the performance is, in general, more than adequate to meet the needs of this application domain. This said, we would like to convince the reviewer that there are no theoretical hurdles to improving performance or scalability, with some additional implementation effort.
> > > > >
> > > > > The approach we would follow would rely on the structure of both subproblems.
> > > > Subproblem 1 (optimize $W$, the factor values for each participant) can be further split into \#rows (participants) of vector optimization problems, which can be solved independently. Similarly Subproblem 2 (optimize $Q$, the loadings over questions for each factor) can be further split into \#rows (questions) of vector optimization problems. This makes it possible to use multithreading to solve multiple vector optimization problems in parallel. Each vector optimization problem in both subproblems is a constrained LASSO problem, as discussed in our previous post, which we solve with ADMM. We empirically found that most of these vector optimization problems converge in less than 10 iterations.
> > > > >
> > > > > Finally, once we trained the factorization model, performing inference on new participants only requires optimizing $W$ and $Z$. However, as $Q$ is fixed during mode inference, the initialization of $W$ and $Z$ is substantially better than that used during training (e.g. we can start with the population average). Therefore, inference is fast for new participants.
> > > >
> > > > - "but this is a generalization of very close methods using ADMM for matrix factorization, and the proofs carry over naturally."
> > > > > We thank the reviewer for elaborating on this point, as we think we can explain our motivation better. We provided the proofs for several reasons:
> > > > > - to establish convergence properties of the proposed algorithm
> > > > > - to determine how one should choose the hyper-parameter $\rho$ (penalty parameter in ADMM)
> > > > > - to demonstrate how one could "generalize" the proof if different constraints were included/replaced
> > > > >
> > > > > There are other ways to split the matrix factorization problem into sub-problems, beyond the ones we used. These splits may induce dissimilar complexity of each subproblem and their corresponding convergence proof. With the extra constraints and penalties, we believe the proof is not a straight generalization, especially given that we also provide a lower bound for the hyper-parameter $\rho$.
> > > > >
> > > > > In the paper, we prioritized using space to introduce our application domain, and motivate the interpretability constraints made possible by our method. While we believe the proofs are important, and a contribution on their own, this is why we opted to present them in abbreviated form with the full details in the Appendix.

---

### Official Review · Reviewer_53sj · 2022-10-28

**Confidence:** 4
**Correctness:** 4
**Technical Novelty And Significance:** 3
**Empirical Novelty And Significance:** 3
**Recommendation:** 8

**Clarity, Quality, Novelty And Reproducibility:**

Clarity

The paper is very clearly written. The formulation of the problem and the optimization procedure are clearly outlined. The paper clearly highlights its novel contributions and makes references to key relevant related work.

Quality

The quality of the paper is sound. The approach is convincingly validated by using synthetic data and by comparing with other similar algorithms. The authors apply their method to real data to estimate latent factors across questionnaires and predict diagnosis, producing compelling results and providing a sufficiently representative view of the real-world applicability of the algorithm. The theoretical section of the paper aimed at demonstrating that the algorithm can converge to a global minima seems solid and there are no errors I could see, although I did not check all the math in detail.

Novelty

The paper introduces a novel factorization method that preserves interpretability. The resulting factorization method is new but the procedures and techniques used to derive it (ADMM, sparsity inducing regularizers and so on) are not. The novelty is somewhat limited but significant nevertheless.

Reproducibility

The authors provide enough information to enable others to reproduce their work. I have not personally attempted to reproduce the key results of the paper.

**Strength And Weaknesses:**

Strengths

A sound method with some theoretical guarantees that performs well in practice and has a wide range of impactful applications.

Weaknesses

**Summary Of The Paper:**

The authors present a novel factorization method for questionnaires that emphasizes preserving interpretability (ICQF). Authors provide theoretical convergence guarantees for their algorithm. The approach is validated using synthetic data with known latent dimensionality. Furthermore, authors apply their approach to real data and show ICQF can uncover a reduced set of common factors spanning multiple questionnaires. This work shows that ICQF algorithm can preserve relevant information for diagnosis and performs well compared with similar approaches.

**Summary Of The Review:**

Overall a solid contribution to the field of factor analysis and a very interesting tool for analysis of questionnaire data. In my opinion this work deserves a spot in the conference.

---

> ### Author Response · Authors · 2022-11-18
> **Detail response of individual comments and questions**
>
> We thank the reviewer for their kind comments, and accurate summary of the paper. We comment at more length on the novelty in the shared reviewer response, as other reviewers also had questions on this point.

---

### Official Review · Reviewer_dWDF · 2022-10-29

**Confidence:** 4
**Clarity, Quality, Novelty And Reproducibility:** The paper is well-written.
**Correctness:** 4
**Technical Novelty And Significance:** 3
**Empirical Novelty And Significance:** 4
**Recommendation:** 8

**Strength And Weaknesses:**

- This is a solid paper. Merits of the proposed methodology have been well substantiated by comparing with alternative methods on data as well as simulation. The choice of cross validation seems appropriate. Theoretical guarantees for the convergence of algorithm have been provided.
- Numerical experiments show significant advantage over existing methods.

Questions:
- For meta-factorization, is it necessary to have equal number of participants $n$ for all questionnaires? Otherwise, it seems one can’t concatenate all factor matrices like in Eq. (6). If so, I wonder how one can relax this constraint? Because in practice, the number of participants may not be the same for each questionnaire due to non-compliance, so requiring the number of participants to be the same could be unrealistic.
- As a follow-up to the last question about meta-factorization, I wonder if the sets of respondents have to be the same for all questionnaires? In practice, they may be different due to non-compliance or missing data. Some participants who answered questionnaire A may not answer questionnaire B.
- About the optimization problem (ICQF), Contribution 2 mentioned: “if this number of factors is close to that underlying the data, the solution will be close to a global minimum”. Could “the number of factor of the underlying data” be more explicitly defined? In practice, I wonder which optimization problem is more non-convex, harder to reach a global minimum and more numerical unstable: the one with number of factors over-estimated or the one with number of factors under-estimated.


**Summary Of The Paper:**

This paper proposes a factorization method for clinical questionnaires by by idea of matrix completion. Given an original data matrix which stacks answers of a questionnaire from all respondents. The method searches by optimizing a regularized loss function under interpretability constraints for a pair of factor and loading matrices so that the product of the factor and loading matrix recover the original matrix. In the situation that there are multiple questionnaires, a meta-factorization is proposed in this paper by first factorizing each questionnaire, and then perform a second factorization on the concatenation of all first-step factor matrices. Numerical experiment has been performed on real questionnaires.

**Summary Of The Review:**

Overall I think it is a high-quality paper. It is well written and its methodology has been carefully tested with numerical experiments and theoretical study. Of course, its technical quality builds on the abundance of literature on matrix factorization and related optimization techniques like ADMM and cross validation, so the methodological novelty is perhaps not its strongest part. However, I feel the method has been well demonstrated with interesting datasets and application, which deserves recognition

---

> ### Author Response · Authors · 2022-11-18
> **Detail response of individual comments and questions**
>
> We would like to thank the reviewer for taking the time to review the paper and raising important questions. In the following, we will address individual comments and questions raised by the reviewer.
>
> - "For meta-factorization, is it necessary to have equal number of participants $n$ for all questionnaires?"
>     > Our apologies if this was unclear. It is not necessary to have the same number of participants across questionnaires. In Figure 5 in Appendix E, we show the availability of participant responses across all 21 questionnaires in the HBN dataset; it can vary considerably.
>
> - "I wonder if the sets of respondents have to be the same for all questionnaires"
>     > Our proposed meta-factorization approach does not require consistent sets of respondents across all questionaires. This was one of our design criteria as, in our experience, there will be both consistently missing questionnaires (e.g. one site contributing participants did not administer a questionnaire) and randomly missing ones (participants never got round to them, or ran out of time).
>     >
>     > Our method produces factors at the first level for each questionnaire, for all participants that have answered it. The resulting factors are concatenated into a single matrix with as many rows as participants, with corresponding mask matrices capturing participant availability. When second level factorization is performed, the values missing in different factors at the first level will be imputed.
>
> - "Could “the number of factor of the underlying data” be more explicitly defined?"
>     > We thank the reviewer for the question, as we now realize this may have been ambiguous. We primarily meant something like the intrinsic dimensionality of the data, if it were being generated as a linear mixing of a number of uncorrelated factors. While there is no ground truth for this, the closest thing for a single questionnaire would be the constructs hypothesized by the designers. There is nothing quite so clearly defined that spans across questionnaires.
>     >
>     > We did carry out recovery experiments using realistic synthetic questionnaire data, with known ground truth and varying amounts of noise, described in Appendix F. These compared the effectiveness of BCV and other procedures for estimating the number of latent factors in a synthetic example, for ICQF, factor analysis and $\ell_1$-NMF. ICQF outperformed them, across all noise levels.
>
> - "I wonder which optimization problem is more non-convex, harder to reach a global minimum and more numerical unstable: the one with number of factors over-estimated or the one with number of factors under-estimated."
>     > Suppose that there is a ground-truth factorization ($W$, $Q$) that reaches a global minimum, we will never be able to reach it if we under-estimate the number of factors (i.e., dimension of $W$). However, the factors captured will be the dominant ones and therefore it is numerically more stable. If we over-estimate the number of factors, it may be unstable in the sense that noise in the training data could be captured as additional factors. Specifically, it will be more sensitive to, e.g., initialization or stopping criteria of the iterative scheme.
>     >
>     > In terms of convexity of the problem, over-estimating the number of factors gives a flatter loss landscape near local minima as observed in the cited paper (Bjorck et al. 2021). One may therefore need a stronger regularization to convexify the problem (equivalent to stronger priors on the factors). Although one may argue that there could be more local minima which are close to global minima, over-estimating number of factors makes it harder to interpret the factorization. In the under-estimation scenario, a weaker regularizer will be sufficient for problem convexification. It is also easier to interpret. However, since the dimension of latent space is insufficient, minor factors may not be captured. Therefore, an accurate estimation of number of factors is essential to maintain both accuracy and interpretability of the factorization model.
>
> Reference:
> - Bjorck, J., Kabra, A., Weinberger, K. Q., & Gomes, C. “Characterizing the Loss Landscape in Non-Negative Matrix Factorization” Proceedings of the AAAI Conference on Artificial Intelligence. Vol. 35. 8, pp. 6768–6776, 2021.

---

### Official Review · Reviewer_988h · 2022-11-01

**Confidence:** 3
**Correctness:** 2
**Technical Novelty And Significance:** 3
**Empirical Novelty And Significance:** 1
**Recommendation:** 5

**Clarity, Quality, Novelty And Reproducibility:**

Please refer to the review section.


**Strength And Weaknesses:**

Please refer to the review section.


**Summary Of The Paper:**

A new matrix factorization method is presented in this paper in order to improve the interpretability of questionnaires through bounds and sparsity constraints. The proposed method utilizes an optimization procedure with theoretical convergence guarantees to detect latent dimensionality in synthetic data. The empirical studies only applied to two datasets, including, a commonly used general-purpose questionnaire, and the Healthy Brain Network study. Overall, I found this paper difficult to follow, and I am not sure from the standpoint of machine learning how the proposed method is a novel one. Additionally, the empirical studies did not provide sufficient evidence to support the proposition that the proposed method is superior to existing approaches. Major concerns and minor comments are presented in the review section.

**Summary Of The Review:**

1) The performance of the proposed method is only benchmarked on two datasets, i.e., the general-purpose questionnaire, and the Healthy Brain Network study. There are several reasons why empirical studies did not provide sufficient evidence for evaluating a novel machine learning approach - such as batch effects, data collection error, missed label handling, etc. Several more datasets should be benchmarked in order to be able to trust the current results.

2) According to the data section, the Healthy Brain Network has 11 class labels. Therefore, reporting ROC-AUC (which is mainly used for binary classification) in Table 1 and Figure 3 may not be an appropriate method of evaluating the proposed method. What was the reason behind not using classification accuracy in the process?

3) The proposed method should be compared with the state-of-the-art factor analysis techniques and method techniques that are widely used for questionnaire/tabular data.

4) According to the author(s), the proposed method is more interpretable than existing approaches. The proposed method could be compared with other techniques in terms of explainable artificial intelligence (XAI).

5) In Figures 1 and 2, why validation errors are reported rather than test errors?

6) In Figure 1 Bottom Right, what is X axis metric? What are the upper and lower bounds for this axis?

7) The current format of Figure 1 Top is not informative. Furthermore, this figure illustrates that the majority of the factors in Q are close to zero - we assume that these characteristics cannot significantly affect the prediction procedure. If we apply thresholding or regularization to push them to zero, what will happen?

---

> ### Author Response · Authors · 2022-11-18
> **Detailed response of individual comments and questions**
>
> We would like to thank the reviewer for taking the time to review the paper and raising important questions. In the following, we will address individual comments and questions raised by the reviewer. Due to the restriction of number of characters, our response is split into different comment posts.
>
> - "Overall, I found this paper difficult to follow, and I am not sure from the standpoint of machine learning how the proposed method is a novel one."
>     > We apologize if the paper was unclear. We address the point of novelty in the shared response to all reviewers, and ask the reviewer to please refer to that.
>
> - "The performance of the proposed method is only benchmarked on two datasets", "Several more datasets should be benchmarked in order to be able to trust the current results."
>     > As discussed in the shared response, the main point of the diagnostic classification experiments was to show that our method does not lose information, even though it imposes several useful constraints. This said, we agree with the reviewer that benchmarking on more datasets would be useful. To that effect, we reproduced all diagnostic classification experiments in the CBCL questionnaire from the Adolescent Brain Cognitive Development (ABCD) study, obtained in a completely different population from the Health Brain Network used in the paper. The results were very similar and are now reported in the paper. For more details, we ask the reviewer to please refer to the shared response.
>
> - "reporting ROC-AUC (which is mainly used for binary classification) in Table 1 and Figure 3 may not be an appropriate method of evaluating the proposed method"
>     > There are two considerations in this case. The first is that performance on each diagnostic label must be considered separately, and hence is a binary classification problem. For each binary problem, we use ROC-AUC, as that is a better measure from the perspective of clinician. This is because they may use different thresholds for detection depending on the purpose of the prediction, e.g. screening vs. an expensive intervention).
>
> - "should be compared with the state-of-the-art factor analysis techniques and method techniques that are widely used for questionnaire/tabular data."
>     > Factor analysis is the most commonly used technique to reduce a questionnaire to a few latent variables representing constructs. In fact, most questionnaires are designed so that this is the case, by having different questions with correlated answers. We combine it with promax rotation (Gaskin and Happell 2014; Goretzko, Pham, and Bühner 2021) and Horn's parallel analysis to determine the optimal number of factors. This goes beyond most studies, which would simply try different numbers of factors specified by the researcher (or a less robust measure such as AIC, which we also tried and abandoned, due to instability). Hence, it **is** the state-of-the-art factor analysis technique.
>     >
>     > Factor analysis was meant to be a baseline familiar to the clinical community designing questionnaires. We added sparse NMF as a second baseline because it is the matrix factorization method that is closest to ours, albeit without any of the interpretability constraints added by our method. Given that one of our key results is demonstrating that those constraints can be imposed without loss of the information needed for diagnostic performance, we feel this is an appropriate baseline.

---

> > ### Author Response · Authors · 2022-11-18
> > **Detailed response of individual comments and questions**
> >
> > - "could be compared with other techniques in terms of explainable artificial intelligence (XAI)"
> >     > Given that, in this instance, we have no ground truth for interpretability, we are not sure of what other techniques (or experiments) would be suitable. We aim for interpretability in two ways. The first is a priori, by imposing constraints that our clinical collaborators tell us would make models more interpretable (e.g. if the factor loadings have to be on the same scale as possible answers, they can be interpreted as a pattern of answers; in factor analysis, they would be an arbitrary linear combination of answers, and thus hard to interpret). The second is post-hoc, by verifying whether the questions associated with a factor make sense to a clinician.
> >     >
> >     > For a single questionnaire, this is captured to some extent by evaluating at whether the questions from the same factor are part of the same "subscale" (a group of questions focusing on a construct). In CBCL, this was the case for our method, where each factor loaded on questions in a single subscale or, at most, two subscales that corresponded to combinations of symptoms observed in patients (this can be seen in Figure 1 top, for every factor, with subscales labelled at the top, e.g. aggressive, attention, etc). In a meta-factorization, there is no notion of subscale. There, the qualitative evaluation is whether the set of important questions for a meta-factor is coherent, even though they come from multiple questionnaires. This was deemed the case for **every** meta-factor by our clinical collaborators (and we provide one example in Figure 2).
> >
> > - "In Figures 1 and 2, why validation errors are reported rather than test errors?"
> >     > We apologize for the lack of clarity here. All the results reported in the paper are obtained on test sets. What is being shown in Figure 1 and 2 are the error curves on the validation set of the block cross-validation procedure, which was used to select both the sparsity parameter $\beta$ and the dimensionality $k$. This is carried out within the training set, with the error pertaining to the subset of it that is used for validation.
> >     The motivation for these subfigures is to demonstrate that the inflection point for showing optimal $(\beta, k)$ setting is obvious using the proposed algorithm, even in a real dataset. This is also aligns with our results on realistic synthetic questionnaire data in Appendix F.
> >     >
> >     > In more detail, in the block cross-validation procedure used in the paper we randomly masked some sub-blocks of the **training** dataset, shuffled row-wise and column-wise. We then perform ICQF to obtained the imputed matrix (via multiplying factors $W$ by their loadings $Q$). The randomly masked sub-blocks act as a validation dataset for the factorization procedure. Notice that $\beta$ and $k$ are hyper-parameters and should only be determined without seeing the **test** data. By comparing the entries of randomly masked sub-blocks from the original matrix and the imputed result, we obtain the reconstruction error. For different settings of $(\beta, k)$, the error reveals whether the hyper-parameters induce over-fitting. For instance, if $k$ is too large, the factors may be capturing some structure in the noise. If $\beta$ is too weak, the sparsity regularization is not strong enough to de-noise the data matrix. As it is very similar to cross-validation model selection, we used "validation error" to designate that reconstruction error.
> >
> > - "In Figure 1 Bottom Right, what is X axis metric? What are the upper and lower bounds for this axis?"
> >     > The bottom right of Figure 1 shows the top-10 questions with the highest loadings for the first factor of $Q$ (i.e. the first row of Figure 1 Top), using a horizontal barplot. The height of each bar (i.e., the $X$-axis) represents the magnitude of the corresponding entries in the first factor of $Q$. As this is a single questionnaire factorization, we simply constrained the bounds to be equivalent to the bounds set for the responses in the questionnaire.
> >     >
> >     > We thank the reviewer for raising this question. The original $X$-axis magnitude was incorrect, as it exceeds the maximum magnitude of a possible answer in the CBCL questionnaire. This was caused by an error in manual production of the figure. The correct version for CBCL factor 1 is shown in Figure 7 in Appendix H of the original manuscript, and has been corrected in the main text of the revised manuscript.

---

> > > ### Author Response · Authors · 2022-11-18
> > > **Detailed response of individual comments and questions**
> > >
> > > - "The current format of Figure 1 Top is not informative. Furthermore, this figure illustrates that the majority of the factors in $Q$ are close to zero -- we assume that these characteristics cannot significantly affect the prediction procedure. If we apply thresholding or regularization to push them to zero, what will happen?"
> > >     > What is shown in Figure 1 top are the loadings of each factor in $Q$ across all the questions in the questionnaire. The motivation for Figure 1 top is to see whether factors correspond neatly to subscales, or combinations thereof, based on their respective question loadings. Essentially, those loadings in $Q$ are what is used when inferring the value of the factor scores $W$ for a new subject. Every factor in $Q$ has high loadings on at least a few questions, as do the confounding variables (young/old, male/female), which act as fixed factors.
> > >     >
> > >     > If we apply a stronger regularization on $Q$, the corresponding factor scores $W$, will be altered. As shown in Figure 1 bottom-left, non-optimal choices of $\beta$ (different curves) will also induce a larger reconstruction error (worse matrix reconstruction) and affect the choice of $k$. Similarly, if we apply thresholding on the trained $Q$, it will also affect the resultant $W$ during model inference on the test set.
> > >     >
> > >     > As the factors $W$ are used as the input features of the logistic classifiers, every factor in $W$ can potentially affect the prediction of a diagnosis. Whether it does also depends on the weight for that factor in the prediction model (e.g. factor 1 might be relevant for borderline personality disorder, but not for depression). If we impose a *much* stronger regularization on $Q$, as mentioned earlier, the optimal choice of $k$ and the resultant factors $W$ will be potentially altered. This may subsequently affect the distribution of high loadings in every factor in $W$, and changes in prediction performance might be significant. If we truncate tiny-valued factor loadings in $Q$, we agree with the reviewer that it will not dramatically change the $W$ that inferred. Likewise, prediction performance will only be slightly altered. However, thresholding would also introduce a new hyper-parameter (the threshold) to the procedure and it would be non-trivial to determine the threshold appropriately.
> > >
> > > Reference:
> > > - Gaskin, C. J., & Happell, B. “On exploratory factor analysis: A review of recent evidence, an assessment of current practice, and recommendations for future use”. International journal of nursing studies 51.3, pp. 511–521, 2014.
> > > - Goretzko, D., Pham, T. T. H., & Bühner, M. “Exploratory factor analysis: Current use, methodological developments and recommendations for good practice”. Current Psychology 40.7, pp. 3510–3521, 2021.

---

### Author Response · Authors · 2022-11-18
**Shared response**

## Shared Response

We would like to thank all reviewers for taking the time to provide thoughtful feedback on our paper. We were pleased to see that several reviewers agree on various positive aspects, namely

- clarity of the writing (dWDF, 53sj, V6jc and R2ij)
- demonstration of the method on interesting datasets and applications (dWDF, 53sj, V6jc)
- solid technical contribution, both empirical and theoretical (dWDF, 53sj)

Some issues have also been raised by multiple reviewers, specifically:

- the method is an incremental improvement on other matrix factorizations (dWDF, 53sj, V6jc, R2ij)
- the method does not greatly outperform baseline factorization methods on diagnostic classification tasks (988h, V6jc, R2ij), and other contributions are unclear (R2ij)

So we will address those here, prior to addressing individual reviewer comments and questions.

### Novelty and motivation of the method

The method was developed **because** we could find no other non-negative factorization method that would enforce the constraints we wanted for factorization of a single questionnaire. The constraints were defined in discussion with our clinical collaborators, to capture characteristics that they felt increased the interpretability of a factorization. A constraint such as "factor loadings must be interpretable as a pattern of answers", for instance, is not found in any of the usual NMF variants. The bounded constraints on $W$ also allow a direct extension of ICQF to meta-ICQF in the multi-questionnaire setting. This, again, came as a result of conversations with our collaborators on identifying common dimensions of psychopathology manifesting across questionnaires.

Adding these constraints is non-trivial, as attested by the derivation of the procedure in Section 3.2 and Appendix A. The components of the procedure are well understood (e.g. ADMM); the novelty is in their composition for this goal. Doing so in a way that allows us to demonstrate convergence of the procedure  -- and near optimality, in certain conditions -- is an important component of proposing a new method. Finally, the handling of confounds or missing data are aspects that are important in practice; factor analysis, commonly used, requires separating the data by levels of the confounds -- with loss of power -- and the use of a separate  method for imputation.

### Quantitative evaluation of performance against baseline methods

As we realize this was unclear for several reviewers, we would like to stress that the use of diagnostic classification from factors as a measure of performance of our method vs others is **not** meant to serve as a competitive benchmark. In our experience, the various factorization methods we have tried will yield similar performance in this task, across diagnoses. Rather, our goal was to show that having all the constraints that favor interpretability comes at **no** cost in terms of diagnostic classification, i.e. our method preserves that information; this is something that our clinical collaborators deeply care about.

### Quantitative evaluation of interpretability

There is no ground truth for interpretability in this instance, which makes it hard to formulate quantitative criteria. The closest, in terms of an individual questionnaire, is the human grouping of the questions into subscales; this doesn't, however, reflect relative importance between questions. This is what drove us to include Figures 1 and 2 to facilitate qualitative assessment of a factor (and there are corresponding tables for every factor, in Appendix H and I). The solutions in the figures should be evaluated in the light of the clinical uses of factor analyses and of the reasons for our choice of constraints, as discussed in the introduction.

We did introduce one quantitative assessment of interpretability: Does the meta-factorization identify the most informative questions, out of around 1000 available across 21 questionnaires? Figure 3 shows that our method significantly outperforms the baseline methods at this task; the same is true across almost every diagnostic classification, as shown in Appendix L. This aspect of interpretability makes it possible to create synthetic questionnaires with just the most informative questions. We apologize if this contribution was unclear, as it is another point that clinical collaborators are enthusiastic about.

---

> ### Author Response · Authors · 2022-11-18
> **New analyses added in response to reviewer comments**
>
> In the light of the questions above, we added two new analyses to the paper.
>
> The first analysis is an assessment of diagnostic prediction performance in a new dataset. In the results reported in Table 1, we used the CBCL questionnaire from the Health Brain Network (HBN) study (now CBCL-HBN). We obtained the CBCL questionnaire from a different study population, the Adolescent Brain Cognitive Development (ABCD) study, consisting of over 11,000 youths aged 9--10 (now CBCL-ABCD).  ABCD has the same diagnostic labels as HBN. We factored the CBCL-ABCD questionnaire using ICQF and the other baseline methods, and carried out the diagnostic classification tasks in exactly the same way as for CBCL-HBN.  The average results for CBCL-ABCD across diagnoses are very similar to those in the original CBCL-HBN.
>
> | **Questionnaires** | ICQF | $\ell_1$-NMF | FA-promax | subscales | raw |
> | --- | :---: | :---: | :---:| :---: | :---: |
> | CBCL-HBN | 0.782 | 0.777 | 0.778 | 0.766 | 0.788 |
> | CBCL-ABCD | 0.776 | 0.752 | 0.771 | 0.767 | 0.786 |
>
> We have included CBCL-ABCD results for each diagnostic prediction problem in Appendix K, as well as a full description of the dataset, and a sentence referring the reader to this within the main text.
>
> The second analysis is an assessment of replicability of factor loadings found on the same questionnaire (CBCL) across different populations (HBN and ABCD). Replicability of factor loadings across populations is one of the criteria used to validate factor models for research or clinical use; having CBCL in two populations makes it possible to do this. For each factorization method, we quantify how similar the resulting factorizations are in terms of the loadings across questions that are learned for each factor, for a fixed number of factors. We measure this by computing the correlation between factor loading vectors in the solutions in each dataset, and greedily matching them. As visible in Figure 15 of Appendix K, ICQF factor loadings are as or more similar across ABCD and HBN than those of baseline methods.

---

### Decision · Program_Chairs · 2023-01-20

**Decision:**

Reject

**Justification For Why Not Higher Score:**

The paper is a standard Nonnegative matrix factorization algorithm.

**Justification For Why Not Lower Score:**

N/A

**Metareview: Summary, Strengths And Weaknesses:**

In this paper, the authors present a procedure for interpretable nonnegative matrix multiplication. Two of the reviewers were positive and, three reviewers were negative. After I read the paper, I believe the positive reviewers provided superficial reviews without properly assessing it.

The loss proposed in Section 3 (bottom of page 3) is a standard square loss with standard nonnegative constraints for W and Q. The regularizers for W and Q are standard too. The only aspects that might be novel are the constraints over Z, not to provide values that are not compatible with M, and the mask \mathcal{M}. But these are minor and straightforward contributions at best. The authors mention that their procedure is interpretable. But they do not say why or what makes their algorithm interpretable.

The results in Section 5.1 for the CHILD BEHAVIOR CHECKLIST look like the ones that we could get with Nonnegative matrix factorization, which are not shown for this problem. Some factors are very clear others not so much. The results in 5.3 in which the proposed results are compared with NMF and other methods show almost no difference in AUC terms, which makes sense.

For this paper to be accepted, there should be an apple-to-apple comparison to a standard NMF for the proposed data in Section 5.1. For example, the authors should replicate Figure 1 with other methods. From this comparison, the authors should explain why their algorithm is better. Also, if the solution is more interpretable than NMF they should indicate what makes it more interpretable. I would also compare with a model that considers the answers are discrete, e.g, http://proceedings.mlr.press/v33/gopalan14.pdf.